



# Passive Acoustic Measurement of Bedload Grain Size Distribution using the Self-Generated Noise

Teodor I. Petrut[1,2], Thomas Geay[1], Cédric Gervaise[4,5], Philippe Belleudy[2], Sebastien Zanker[3]

[1] Université Grenoble Alpes, Grenoble INP, CNRS, GIPSA-Lab, Grenoble, France
[2] Université Grenoble Alpes, CNRS, IRD, Grenoble INP, IGE, Grenoble, France
[3] Électricité de France, DTG division, Grenoble, 38040, France
[4] Institut de recherche CHORUS, Phelma Campus, 3 Parvis Néel, 38000 Grenoble, France
[5] Chaire de recherche CHORUS, Fondation partenariale Grenoble INP, 46 avenue Felix Viallet, 38000, Grenoble, France

*Correspondence to*: Thomas Geay (th.geay@gmail.com)

**Abstract.** Monitoring sediment transport processes in rivers is of particular interest to engineers and scientists to assess the stability of rivers and hydraulic structures. Various methods for sediment transport processes description were proposed using conventional or surrogate measurement techniques. This paper addresses the topic of the passive acoustic monitoring of bedload transport in rivers and especially the estimation of the bedload grain size distribution from self-generated noise. It
discusses the feasibility of linking the acoustic signal spectrum shape to bedload-grain sizes involved in elastic impacts with the bed river treated as a massive slab. Bedload grain size distribution is estimated by a regularized algebraic inversion scheme fed with the power spectrum density of river noise estimated from one hydrophone. The inversion methodology relies upon a physical model which predicts the acoustic field generated by the collision between rigid bodies. Here it is proposed an analytic model of the acoustic power spectrum generated by the impacts between a sphere and a slab. The proposed model is written
as linear system of analytic power spectra weighted by the grain size distribution. The algebraic system of equations is then solved by least square optimization and solution regularization methods. The result of inversion leads directly to the estimation of the bedload grain size distribution. The inversion method was applied on real acoustic data from passive acoustics experiments realized on the Isère River, in France. The inversion of in situ measured spectra reveals good estimations of grain size distribution, fairly close to what was estimated by physical sampling instruments. These results illustrate the potential of
the hydrophone technique to be used as a standalone method that could ensures high spatial and temporal resolution measurements for sediment transport in rivers.

## 1   Introduction

Sediment transport analysis in river catchments are one of the key activities stipulated by the European water framework directive (WFD, 2007) and also applied in French environmental policies. Climate changes and anthropological actions impact
the sediment transport in rivers such that it produces changes in the river morphology and may put at risks ecosystems and hydraulic structures, eventually. One of the major concerns of sediment transport in rivers is determining the total discharge





of bedload transport (Gray et al., 2010). Bedload transport models are highly sensitive to incipient motion which is directly related to river bed grain size distribution (GSD). Bedload GSD is linked to both surface and substrate GSD. In his paper, Parker constructed a two size fraction transport model, assuming that the bedload GSD is identical to substrate GSD, for stable armored bed rivers, and becomes identical to surface GSD whenever the armor is destroyed (Parker, 1990). The development

of surface-based and mixed-size transport models concerned many scientists (Wilcock & McArdell, 1993). Knowing the bedload GSD solves the problem of initiation of motion and, therefore, enhance the accuracy of transport rate prediction. Therefore, measuring bedload leads not only to transport rates but also to bedload GSD to calibrate models. However, obtaining bedload samples during exceptional hydraulic events may be difficult by using traditional bedload sampling techniques (e.g., pressure-difference samplers). To measure a wide range of discharge flows, the scientific community has been interested in

developing indirect, or surrogate, methods that achieve continuous measurements no matter the hydraulic conditions. This paper is dedicated to the monitoring of bedload GSD using the acoustic noise naturally generated by bedload transport in rivers.

Acoustics surrogate methods are divided in two categories: active and passive methods (Gray et al., 2010). Examples of active methods are the acoustic Doppler current profile, aDcp (Rennie & Millar, 2004), or the acoustic mapping velocity

technique (Muste et al., 2016). Active methods use emissions of well-known signals but, actually, in the author's knowledge, no active instrument was conceived to estimate bedload GSD. Besides, the major problem of the active instruments is that they do not properly behave during high flow discharges. This is why the passive instruments are preferred instead of the former. These instruments use seismic or acoustic signals generated by bedload particle impacts. Recorded signals bear information on both sediments impact rate and bedload particles sizes. One of the most used technique consist in recording the signal of

particle impacts on steel objects like plates (Barrière et al., 2015; Rickenmann et al., 2014), pipes (Mao et al., 2016; Mizuyama et al., 2010) or column pipes (Papanicolaou et al., 2009). Other passive instruments consist in directly recording bedload Self-Generated Noise (SGN) by using passive acoustic monitoring (PAM)  (Barton et al., 2006b; Bedeus & Ivicsis, 1963; Geay, 2013, 2017; Thorne, 1986) or seismic monitoring (Gimbert et al., 2014; Roth et al., 2016; Tsai, et al., 2012). Measuring bedload GSD with passive methods has been achieved using plates (Barrière et al., 2015; Krein et al., 2014; Rickenmann et al., 2014)

or pipes (Dell'Agnese et al., 2014; Mizuyama et al., 2010; Papanicolaou et al., 2009), and SGN (Johnson & Muir, 1969; Jonys, 1976; Thorne, 1986; Geay, 2017), by using experimental laws of calibration. Concerning seismic methods, bedload GSD measurements were not yet proposed as a direct application.

The existence of a link between the GSD and the features of vibrational signals has been demonstrated in several experiments (Belleudy et al., 2010; Bogen & Møen, 2001; Krein et al., 2008; Turowski & Rickenmann, 2011). By coupling

geophones with steel plates (Barrière et al., 2015; Wyss et al., 2016) produced composite power laws by linking both peak amplitude and peak frequency to the grain size. Using the Japanese pipe, (Mao et al., 2016) proposed an empirical model based on multi-channel recorded amplitude ratios to estimate different percentiles of grain diameters ($D_{16}$, $D_{50}$ and $D_{84}$). The only metric exploited in this kind of measurements is the amplitude of shocks on steel structures. Thus, these passive techniques involving shocks on steel structures offer high quality signal, or signal-to-noise ratios (SNR). The analyzed physics is the same





as in the case SGN measurements by PAM, which is the rigid body radiation caused by hertzian impacts between sediments. In the case of SGN measurements, unlike the steel structure impacts measurements, the SGN signal amplitudes are not usable for grain size inversion because of the issues concerning the sound propagation throughout the reach (the amplitudes depend on the distance between the shocks and the hydrophone). This makes the amplitude a futile metric to infer grain-size information from SGN signals.

Several studies in the field highlighted that the frequency content (i.e., spectrum shape) of SGN signals is heavily dominated by grain sizes. For example, (Jonys, 1976) showed by laboratory experiments with ceramic spheres that spectral peak frequency is linked to sphere diameter. The author found a peak frequency at about 1 kHz for 75 mm-diameter particles, at 2.2 kHz for the 38 mm-diameter and 4 kHz for 19 mm-diameter. This means that a doubling of grain size is almost equivalent to halving of peak frequency. Extensive researches on GSD estimation by SGN recordings were made by (Thorne, 1986) where he presented two strategies for inversion of acoustic spectra to estimate GSD. Results were encouraging as GSD were roughly estimated. These techniques are based on experimental measurements that have been made in a rotating drum with specific conditions that are different from the conditions found in rivers (e.g. impact velocities, acoustic propagation). Besides, his inversion techniques raise issues because of the broadband nature (shape) of spectra, even for uniform sediments. The author himself assumed that this was the major cause for inaccurate estimations of GSD from composite spectra.

This paper proposes an inversion method that solves the issue of spectrum shape and which accurately estimates the entire bedload GSD curve. This proposed method is conceived to be transferable to a large set of operational contexts. The procedure of inversion is based on a physical direct model which is presented in the first part of this paper. In the second part, the inversion algorithm is presented in the form of a technique for solving least square (LS) problems with a regularization condition about the positivity of GSD curve. Simulated acoustic spectra and their inversion are used to test the robustness of LS methods to measurement uncertainties. In the third part, the LS inversion algorithm is applied on field measurements done in the large gravel-bed Isère River, France. GSD estimated with our method are compared to GSD measured with a pressure-difference sampler. Additionally, the cross-sectional variability of bedload GSD is analyzed using both, acoustic and direct, measurements. Finally, results are discussed to give a technical overview on the proposed inversion method.

## 2 SGN model

### 2.1 Analytic model of Hertzian impact between a sphere and a slab

This section deals with theoretically modeling of impact between a sphere and a slab (Akay & Hodgson, 1978; Hunter, 1957), as the main assumption of this study is that the acoustics of gravel is described by impacts between bedload sediments and the bed river. To prove the feasibility of this assumption, the study includes some comparative facts with the sphere-sphere impact model (Koss & Alfredson, 1973; Thorne & Foden, 1988).

The collision between bed particles radiates energy, such a rigid body radiation phenomenon is due to both vibrations and accelerations. These processes are very well separated with respect with their dominant frequencies, such that the spherical




mode vibrations generates much higher frequencies than the acceleration-based sound (Barton, 2006a; Thorne & Foden, 1988). The acoustic effect of accelerating rigid bodies is mathematically modeled by (Kirchhoff, 1883). A framework was constructed by (Hertz, 1882, Goldsmith, 2003; Hunter, 1957) to mathematically model acceleration profiles from elastic impacts between two solid rigid bodies like two spheres or a sphere and a slab. Mathematically, the acoustic pressure field generated from the acceleration of a rigid body is evaluated by the integral convolution from Eq. (1) (Koss & Alfredson, 1973; Akay & Hodgson, 1978; Thorne & Foden, 1988). The integral consists of the convolution between the Kirchhoff's impulse response $p_I$ and an acceleration profile $A$. In the case of elastic (hertzian) impacts the acceleration occurs during the impact and so integral is evaluated by intervals with respect to contact duration $T_c$. The contact duration $T_c$ is modeled by Hertz's law and it is put in a simplified form in Eq. (2), for both sphere-sphere and sphere-slab impact models.

$$p(t) = \int_0^\chi p_I(t) \cdot A(t-\tau)d\tau \tag{1}$$

$$T_c = \vartheta^{(1)}\left(\zeta\rho_s\right)^{0.4} aU^{-0.2} \tag{2}$$

where $t$ is the temporal variable and $\chi = t$, if $0 \leq t' \leq T_c$, $t'$ is the delayed time due to sphere geometry, $t' = t - (r - a)/c$, $r$ – the distance of measurement of the sound from the contact point, see Fig. 1a-b, $a$ – the radius of sphere, $c$ – the sound celerity, $\chi = T_c$, if $t' > T_c$, and $\rho_s$ – material density and $U$ – the impact velocity. In Eq. (2), the constant $\vartheta^{(1)} = 9.229$, for the sphere-sphere impact where the spheres' radii are equal, and $\vartheta^{(1)} = 10.601$, for the impact between sphere and slab (considered here). The material parameter $\zeta = (1-v^2)/(\pi E)$, where $E$ – Young's modulus, $v$ – Poisson ratio. The general form of acceleration profile is provided by (Goldsmith, 2003) and it is rewritten in a more convenient form for both impact models in Eq. (3).

$$A(t) = \begin{cases} \vartheta^{(2)} \dfrac{U}{T_c}\sin\left(\pi\dfrac{t}{T_c}\right), & 0 \leq t \leq T_c \\ 0, & t > T_c \end{cases} \tag{3}$$

where the constant $\vartheta^{(2)} = 1.5708$, for sphere-sphere impact and $\vartheta^{(2)} = 3.353$, for sphere-slab impact.

The first important observation from the Eq. (3) is the half-period sinusoidal form of the hertzian acceleration. The two modeled acceleration laws show close frequencies as the constant $\vartheta^{(1)}$ from $T_c$'s formula is not dramatically different from one case to another. If the frequency of acceleration of sphere-sphere impact is 1000 Hz, then the frequency of acceleration of sphere-slab impact is 909 Hz which is almost only 10% of deviation. The maximum amplitude of acceleration for the impact between two spheres of radius $a$ is almost two times less than the impact between sphere of radius $a$ and a slab, considering the same $U$ and $T_c$.

The integral convolution in Eq. (1) is transformed into multiplication in the complex Fourier space. Thus, analytical magnitude spectrum of the noise from the rigid body acceleration, $F_{acc}$, is given in Eq. (4a).

$$F_{acc}(\omega) = F(p_I(t)) \cdot F(A(t)) \tag{4a}$$

where

$F(p_I)$– defined in Eq. (4b), is the FT of Kirchhoff's impulse response $p_I$ (Koss & Alfredson, 1974), for a sphere of radius $a$,



F(A) – defined in Eq. (4c), the FT of hertzian acceleration due to elastic impact between two same size and same material spheres, or between same material sphere and slab, and ω – angular frequency.

$$F(p_I) = \frac{\rho a^3 c}{r^2}(c + j\omega r)\frac{2c^2 - (\omega a)^2 - j2\omega ac}{\left[2c^2 - (\omega a)^2\right]^2 + (2\omega ac)^2}\cos\theta \tag{4b}$$

$$F(A) = \vartheta^{(3)}\frac{U}{\pi^2 - (\omega T_c)^2}\left(e^{-j\omega T_c} + 1\right) \tag{4c}$$

$\omega$ – angular frequency vector, $\omega = 2\pi f$, $j$ – imaginary unit and $\vartheta^{(3)} = \pm\pi^2/2$ for sphere-sphere impact and $\vartheta^{(3)} = 1.067\pi^2$ for sphere-slab impact.

The case of sphere-slab impact is treated below. The hard reflector boundary condition permits to the modeler to consider the impactee the mirrored sphere with respect to the slab's surface, as illustrated in Fig. 1b. Therefore, the total spectrum of the acoustic pressure field is modeled as the subtraction of the two individual spectra, or the addition if it is considered the

10 sign change in the case of mirrored sphere, as it is given in Eq. (5). Besides, the phase spectra will be changed according to the time delay $T_d$ of sound arrival due to distance of measurement and sphere's geometry.

$$F_{im}(\omega) = F_{acc}(\omega) - F_{acc}(\omega) \cdot e^{-j\omega Td} \tag{5}$$

Introducing Eq. (2), (3) and (4a-c) in Eq. (5), one obtains the complex magnitude spectrum of the impact between a sphere and a slab. The spectrum contains complex numbers so one applies the multiplication between the spectrum and its conjugate, to find the magnitudes of power spectrum, Eq. (6).

$$|F_{im}|^2 = F_{im} \cdot F_{im}^* \tag{6}$$

where $F_{im}^*$ is the complex conjugate of $F_{im}$.

As the present research wants to inverse power spectrum densities (PSD), the powers $|F_{im}|^2$ from Eq. (6) will be accordingly scaled, as seen in Eq. (7). This is the analytical model used in this paper to inverse acoustic spectra.

.

$$P(\omega) = \frac{2}{\frac{N_{sig}}{F_s} \cdot S} \cdot |F_{im}|^2 \tag{7}$$

where the units are in (Pa$^2$Hz$^{-1}$), $S = \frac{1}{N_{sig}}\sum_{i=1}^{N_{sig}}w_i^2$ , $w_i$– coefficients of the weighting window (*Blackman* for the kind of signals recorded in the field), $N_{sig}$ – the number of samples of the impact signal and $F_s$ – the sampling frequency. The impulse duration, $N_{sig}/Fs$, takes into account variability with dimension of the impacting sphere and it is at least 5 milliseconds, in the case of

25 the smallest modeled spheres, of 1-mm diameter.





An example of analytical waveform computed using the Akay's analytical model (Eq. 7 from (Akay & Hodgson, 1978)) is presented in the Fig. 2a. The impacting sphere has 20 mm in diameter, the material is granite and the impacting velocity is 1 m s$^{-1}$. The shape of waveform is approximately one and a half period sinusoid. It is observed that the difference of two pressure fields is in fact a sum, between the impacting waveform (rarefaction part) and the reflected waveform (compression part), due to the fact that the hard bottom returns the same-amplitude waveform with changed sign. This is the same as in the case of sphere-sphere impact, where one sphere decelerates whereas the other accelerates. Therefore, the individual generated pressure fields will have opposite signs. The power spectrum density modeled by Eq. (7) is shown in Fig. 2b. Here, the spectrum has a principal lobe and numerous side lobes. The principal lobe has the peak at the frequency of approximatively $1/(1.1 \cdot T_c)$ and the side lobes are approximately associated with the term $\cos(\omega T_c)$, also observed by (Thorne & Foden, 1988).

In Fig. 2c it is shown that the frequency peaks of spectra from both types of impact model are decreasing with the sphere's diameters (from 1 mm to 150 mm) as experimentally observed in (Thorne, 1986). Frequency peak as function of diameter, in the case of sphere-slab impact, $f_{peak}(D)=a \cdot D^b$, is given in the case of three impact velocities, $U = \{0.01; 0.1; 1\}$ ms$^{-1}$. The exponents of the regression laws proves the inverse proportionality between $f_{peak}$ and $D$. Besides, the power peaks and peak frequencies increase, for a certain diameter, when the impact velocity increases. There is only a doubling of $f_{peak}$ when $U$ changes by an order of magnitude. This is also proved by the formula of Eq. (2) of $T_c$ (almost the reciprocal of $f_{peak}$) where the parameter $U$ is raised to a weak exponent of 0.2.

The $f_{peak}$ in the case of sphere-sphere impact, modeled for impact velocity $U = 1$ ms$^{-1}$, is higher than in the case of the sphere-slab impact. Here, the analytical model of sphere-sphere spectrum was computed using the Eq. (4a) and (5), with the two pressure fields auditioned instead subtracted. This gives the same results as the spectral model from the paper (Thorne & Foden, 1988)). To give an idea, a 90 mm-diameter particles in sphere-sphere impact has spectrum $f_{peak} = 2650$ Hz, see detail of Fig. (2c), whereas sphere-slab impact has $f_{peak} = 2350$ Hz, so the 300 Hz represents circa 15% of variation from one scenario to another.

It is worth to mention that $U$ greatly influences the power peak, if the former is changed by one order of magnitude. On the other hand, the power peak of the sphere-sphere impact is slightly weaker than the sphere-slab. This could justify the reason to use the sphere-slab impact physics instead of sphere-sphere impact, to reduce eliminate the need for the dimensions of the impactee object.

## 2.2    PSD model of the SGN generated by a mixture of sediments

In the previous section, the analytic PSD was defined for the impact between two solid rigid objects, like two spheres or a sphere and a slab, representing the sediments and the bed river in reality. In this paper, the focus will be on the sphere-slab impact. In this section, we model the PSD of a sediment mixture using these analytic PSD. Assuming that particle collisions are random and independent noise sources, the model of total PSD can be expressed as a linear summation of the elementary spectra (Johnson & Muir, 1969; Jonys, 1976; Thorne, 2014). The acoustic bedload model under discussion is defined in the scalar form in Eq. (8) and the matrix form in Eq. (9).





$$P_{P\Sigma\Delta} = \sum_{i=1}^{K} d_i \cdot n_i \tag{8}$$

$$\begin{pmatrix} d_{1,1} & d_{2,1} & \cdots & d_{K,1} \\ d_{2,1} & d_{2,2} & \cdots & d_{K,2} \\ \vdots & \vdots & \ddots & \vdots \\ d_{N_{FFT},1} & d_{2,N_{FFT}} & \cdots & d_{K,N_{FFT}} \end{pmatrix} \cdot \begin{pmatrix} n_1 \\ n_2 \\ \vdots \\ n_K \end{pmatrix} = \begin{pmatrix} P_1 \\ P_2 \\ \vdots \\ P_{N_{FFT}} \end{pmatrix} \tag{9}$$

$$\underbrace{\qquad\qquad\qquad\qquad}_{\Delta=\{P_i(f)|\overline{i=1,K}\}\ (Eq.\,7)} \quad \underbrace{\quad}_{\Gamma_{PMF}} \quad \underbrace{\quad}_{P_{P\Sigma\Delta}}$$

where $P_{P\Sigma\Delta}$ is the PSD of the transported sediment mixture, of dimension $N_{FFT}$, $d_i$ are power values from the spectrum $P_i(\omega)$ of the impact of spheres of diameters from size class $i$, Eq. (7) . Thus, each column of the dictionary $\Delta$ is an analytic PSD of one impact between a sphere of diameter $D_i$ and the slab. The $n_i$ is the weight the size class $i$ of the total spectrum $P_{P\Sigma\Delta}$, or put it in another words, $n$ is proportional to a probability mass function (PMF) of the number of collisions, noted by $\Gamma_{PMF}$. The index $i$ takes integer values, from the lowest limit, 1 mm, to the highest one, $K$ mm, where $K$ is the largest diameter considered in modelling. Here, the studies considers $K$ equal to 150 mm.

The random variable of the GSD is the number of collisions $N$, and so the complete notation is $\Gamma_{PMF}(N = n_i)$. However, one needs to transforms this variable to the weight (mass) of sediments $M$, to facilitate the comparison with the measured GSD by bedload samplers. Thus, the variable $N$ from $\Gamma_{PMF}(N = n_i)$ will be multiplied by $D_i^3$ , Eq. (10a) becoming $\Gamma_{PMF}(M = m_i)$. Furthermore, the solution is written as a cumulative distribution form, $\Gamma_{GSD}(M \leq m_i)$, expressing the mass percentage of sediments finer than $D_i$, as in Eq. (10b).

$$m_i \propto n_i \cdot D_i^3, \quad i = 1...K \tag{10a}$$

$$\Gamma_{GSD}(M \leq m_i) \propto \sum \left\langle \Gamma_{PMF_i}(N = n_i) \cdot D_i^3 \right\rangle \tag{10b}$$

This section gave a formal definition to the PSD of a bedload size mixture defined by its GSD. The proportions are considered to be a probability mass function (PMF) of the number of collisions. The size classes concerned in this study are integer numbers, from 1 to $K$, with the narrow size classes of 1 mm.

## 2.3 Global Sensitivity Analysis of the spectrum generated by a mixture of sediments

This analysis was done to determine the importance of input parameters on the shape of the PSD modeled with Eq. (9). The parameters are defined in Table 1. One is interested about the statistical variation of the sphere-slab spectral model, where the inputs are randomly sampled from uniform distributions and the output is the peak frequency $f_{peak}$. Global Sensitivity Analysis (GSA) is made to assess the impact of input parameters on the model output. The Fourier Amplitude Sensitivity Test (FAST) tool for GSA was developed by (Cukier et al., 1973). It is a Monte Carlo method designed to compute the first order indices of sensitivity $S_i$. In such a complex problem, the FAST tool can be used to simplify the model because it helps in determining





which input parameter could be negligible by only looking at its value of $S_i$. The FAST algorithm computes the values of variances from integrals of Fourier coefficients of model outputs instead of computing the variances from n-dimensional integrals, where n is the number of input parameters. In the present GSA one uses the coded version of FAST of (Cannavó, 2012).

The flowchart of the GSA is presented in the Fig. 3. The GSA consist in generating a large number of log-normal GSD by randomly sampling of means and standard deviations, i.e. ln($D$) sampled from uniform distribution from 1 to 150 m, and, respectively, $\sigma$, sampled from uniform distribution from 0.01 to 10. It is important to mention that in this case, ln(D) coincide the $D_{50}$ is the modeled log-normal GSD. All other parameters needed in to compute Eq. (7) and Eq. (9) are given in Table 1. As the output model analyzed is the peak frequency $f_{peak}$ of the simulated PSD curves, the analysis does not claim to completely

describe the model but rather decent idea could be concluded on the model's behavior.

The $2 \cdot 10^5$ Monte Carlo samplings of $f_{peak}$ values are enough to attain stationary values of sensitivity indices ($S_i$), which are presented for each input parameter in Table 2. As it is observed in the table, the standard deviation $\sigma$ of log-normal GSD has the greatest influence on the PSD shape. This is normal $\sigma$ variation affects the values of all percentiles in the GSD. The percentile $D_{50}$ is two times less important than $\sigma$ in affecting the $f_{peak}$. The third greatest parameter as degree of influence on

output is the impact velocity $U$. Back in the Fig. 2(c), it was observed that particles of the same size will generate impact spectra whose $f_{peak}$ doubles as $U$ increases ten times, a property that is validated by GSA, where U is 12% important in the output model. Another relatively influent parameter is the Young's modulus, which means that the type of material also plays a role on the dynamics of the PSD. The distance of measurement is also important, as it the phase shift $T_d$ used in the addition of two coherent acoustic fields and also the Poisson's ratio plays a decent role in this variation. The rest of parameters are

considered of little influence on the values of $f_{peak}$.

As a remark, the first order GSA methodology has the great utility that it could rewrite the Eq. (9) in the form of an empirical law, like the one in Eq. (11), where each variables is normalized between 0 and 1. The computation of high order analysis could be made by Sobol's analysis (Sobol, 2001), but this type of analysis is beyond the scope of this article.

$$f_{peak} \propto \sigma^{0.41} + D_{50}^{0.21} + U^{0.13} + \left(\left(\rho_s, v, E\right)_{material}\right)^{0.14} + r^{0.06} + \theta^{0.0097} + \left(\left(\rho, c\right)_{water}\right)^{0.0096} \tag{11}$$

In conclusion, the first order global sensitivity analysis on the peak frequency modeled by the Eq. (9) shows a comprehensive view on its dynamics with input parameter variation. It is found that the peak frequency is the mainly affected by two parameters, the distribution's standard deviation and the median diameter which together makes out over 60% of output's variation, whereas the impact velocity $U$ and the material of sediments are equally important, with 12 %. In conclusion, the acoustic model is quite complex and care must be taken regarding the recording of the power spectra on the field, as their

shape heavily affect the estimation of GSD. Also, the impact velocity is regarded as a minor factor of uncertainty and because it is almost impossible to be measured for each grain size class, the 12% uncertainty on peak frequency is almost unavoidable.



## 2.4 Hypothesis on the proposed PSD model of mixed impacts

Modeling of single impacts requires definition of parameters typical for river environment, in Table 1. Using the global sensitivity model, it has been shown that PSD shapes are essentially influenced by four parameters: the shape of GSD curve, the median diameters of the colliding particles, the impact velocities and the material. Grain sizes are estimated later thanks to the inversion algorithm presented in the Sect. 3. Concerning the other model parameters, as they are not affecting the PSD shape, they will be fixed for the inversion process, using realistic values. These parameters are listed in the third column of Table 1. The main hypothesis of SGN spectrum model are the following:

    I.    The geometry of the channel and of the material: the river bed is considered as a massive slab and moving particles are considered as spherical.

    II.    Sediment transport assumptions: impact velocities are assumed to be invariant with grain size. This assumption is supported by the relative size effects on bedload transport (Einstein, 1950; Recking, 2016; Wilcock & McArdell, 1993) referring to mobility of finer and coarser particles.

    III.    Acoustic propagation:

- as the bedload GSD is assumed to be homogeneous everywhere in the space, the propagation effects like the attenuation with distance (geometrical spreading models) will not impact the spectrum shape;

- the attenuation due to diffraction from bed and water surface roughness or from the suspended sediments is not considered.

## 3 Inverse model to estimate GSD of bedload particle

The inversion uses LS optimization methods to compute the inverse of dictionary $\Delta$. Normally $K < N_{FFT}$, so $\Delta$ is a non-square matrix. Moreover, the matrix $\Delta$ is possibly rank deficient because the spectra generated by impacts of coarser show very similar shapes, that is, the coarser the particle, the more similar is the produced sound. As a consequence, the pseudo-inverse algorithm is used here to solve the algebraic system of Eq. (9) using least square optimization techniques. The optimization problem is given is defined as the normal equation in Eq. (12). The solution to this problem is the PMF of number of impacts. The estimated PMF is further transformed into the final GSD of mass of sediments, Eq. (10a-b).

$$\hat{\Gamma}_{PMF}(N = n_i) = \arg\min_{\Gamma_{PMF}} \left\langle \left\| \Delta^+ \cdot P_{P\Sigma\Delta} - \Gamma_{PMF} \right\|_2^2 \right\rangle, \quad i = 1...K \tag{12}$$

where $\Delta^+ = \left( \Delta^t \cdot \Delta \right)^{-1} \cdot \Delta^t$ is the pseudo-inverse, $\Delta^t$ means the transpose of matrix $\Delta$, and $\| \ \|_2$ is the L2-norm, i.e. the Euclidian distance.



## 3.1    Numerical test of the LS method

A simulation case is proposed here to test the robustness of the LS inverse method. The simulated PMF is the uniform distribution between 10 and 50 mm and it is expressed in terms of mass of sediments, as it is measured by bedload samplers. The uniform distribution means that 1 kg of $D = 10$ mm has the same probability of producing impact noise as 1 kg of $D = 11$

5    mm, and so on. The chosen PMF is converted back to number of collisions by dividing by $D^3$, to obtain $\Gamma_{PMF}$. Using an impact velocity of 1 m·s$^{-1}$ and the other input parameters defined in Table 1, the simulated PSD is shown in Fig. 4a. The analytic PSD are computed by Eq. (7) for the sphere-slab impacts. The dictionary $\Delta$ contains spectra 1 mm to 150 mm (with size classes of 1 mm the width).

The solution found from the LS pseudoinverse algorithm, Eq. (12), is expressed in terms of PMF of mass of sediments

$\hat{\Gamma}_{PMF_i}(M = m_i)$, $m_i \propto n_i \cdot D_i^3$, $i = 1...K$, and it is shown in Fig. 4b. The solution is smoothed by averaging with a moving window of width 3 mm (equivalent to 3 grain size classes). High instabilities are observed, as the input velocity in simulation ($U = 1$ ms$^{-1}$) is largely different from the one used to model the dictionary $\Delta$ simulation ($U = 0.1$ ms$^{-1}$). This is also eased by the fact that there is a high similarity between individual spectra $d_i$ of close size classes, especially for the largest classes. Thus, the matrix $\Delta$ is ill-conditioned and the problems is ill-posed. Ill-conditioning is linked to the high condition number of the normal

matrix ($\Delta^t \cdot \Delta$). It is defined as the ratio between the largest and smallest eigenvalues of a matrix. A well-conditioned algebraic system requires that the normal matrix should have a condition number as close as possible to 1 (Strang, 2009). In these tests, $\Delta$'s condition number reaches huge values on the order of $10^{12}$-$10^{20}$. In consequence, the similar spectra from the matrix $\Delta$ produce high instability in solution.

To avoid the instability in the LS solution, the Non-Negative Least Squares (NNLS) algorithm (Lawson, 1995) is proposed

to solve the LS problem. This optimization algorithm, Eq. (13), casts non-negative constraints on solution $\Gamma_{PMF}$. The non-negative factorization is widely used, for example, in various domains like image processing or chemometrics. The side-effect of using this algorithm is the strong regularization of solution. The regularization aims to keep the sum of components in $\Gamma_{PMF}$ constant. The solution of the NNLS algorithm, Fig. 4b, show that the instabilities are completely removed off. Besides, it is important to note that the estimated grain size classes are inside the simulated interval of size classes.

$$\hat{\Gamma}_{PMF}(N = n_i) = \underset{\Gamma_{PMF} \geq 0}{\arg\min} \left\langle \left\| \Delta^+ \cdot P_{P\Sigma\Delta} - \Gamma_{PMF} \right\|_2^2 \right\rangle \qquad (13)$$

## 3.2    Robustness of the NNLS algorithm to PSD noise

A particular concern in the theory of statistical signal processing is the variance of computed PSD. In our work, Short-Time Fourier Transform has been used. Fourier transforms are applied on small temporal windows of signal (with an overlap of 50 %). These collections of local spectra are averaged over predefined frequency bins which are narrowband (Oppenheim, 2011).

If the signal is long enough then the averaging of a great number of local spectra diminishes the spectrum variance. In the same time a good spectral resolution is achieved and a great number of size classes may be correctly inferred from the PSD.





The NNLS algorithm will be tested on three simulated spectra which have different degrees of variance. The simulated $\Gamma_{PMF}$ used is identical as the one from the previous section. The simulated signal is obtained by creating a time series of a random signal having the noise-free PSD, to do we put a realization of a white noise in a filter whose transfer function is the noise-free PSD (i.e. the spectrum presented in Fig. 4a). The simulated noised PSD is shown in Fig. 4c. The results of the inversion using the NNLS algorithm show that, even for the worst scenario of variance on spectrum, the inversion method correctly reconstructs the simulated GSD. However, increasing the noise of the PSD requests heavy solution regularization observed by the bump on the curve between 20 mm and 40 mm-size classes.

Finally, it is concluded that the NNLS algorithm is robust with respect to PSD noise and fits to this kind of inversion problem. The inversion procedure will now be tested on in situ measurements.

## 4        Application to real data

### 4.1        Isère River and experimental setup

The Isère River is a piedmont gravel-bed river located in the southeastern France, and it is one of the main tributaries of the Rhône River, which reaches the Mediterranean Sea. The monitoring section is located upstream the city of Grenoble (45° 11'52.8" N, 5°46'14.88"E) see Fig. 5a. In this reach, the mean slope is about 0.06 %, the area of the watershed is 5500 km$^2$ and the average flow rate is 180 m$^3$·s$^{-1}$. At the time of experiments, the 29-30 June 2016, the monitored discharge was on average 300 m$^3$·s$^{-1}$. The measurement section has a rifle-pool morphology with riprap-protected embankments. Two different types of instrument were used: SGN measurements using hydrophone and direct sampling using a pressure-difference sampler, shown in Fig. 5b. All these measurements were carried out from a suspension bridge, Fig. 5c.

### 4.1.1        SGN measurements

SGN measurements were made using a HTI99 hydrophone (High Tech, Inc, *http://www.hightechincusa.com/*) with a sensibility of -160 dB re 1 V μPa$^{-1}$ ±3 dB from 10 Hz to 125 kHz. The hydrophone was connected to an autonomous-waterproof autonomous recorder SDA14 (RTSYS©, *http://www.rtsys.eu/fr/*). The gain of the recorder was set to 15 dB. Signals were sampled at a 312 kHz frequency with a resolution of 24 bits and saved as wav files. The scope of these field experiment was to trace maps of the SGN on the local reach. The hydrophone and the recorder were attached to a free floating river-board. The hydrophone position was about 1 m below the water surface and 1.5 m in average above the bed river. The SGN map consists in launching 12 drift measurements from the bridge which are located thanks to a GPS device connected to the acoustic recorder. Each drift consists of recordings of about 30 to 40 seconds, or in terms of distance, between 50 and 100 m. The river board positions during the drifts are shown in Fig. 6a. The recorded signals were processed to compute acoustic spectra. The 12 acoustic spectra recorded across the river (Fig. 6b) are inversed to estimate the bedload. The river cross-section is of about 60 m. Also, the 12 drift measurements are synchronized with GPS data to compute the SGN map in terms of sound pressure





level (SPL), as it is shown Fig. 6c. The variability of SGN noise from left to right bank can be observed from both spectra and SGN map.

### 4.1.2    Definition of SGN spectrum

SGN signals are measurements of bedload transport noise propagating in the river environment. Several representations of the acoustic signal are presented hereby, computed on the signal recorded in the middle of the river Isère ($X = 34$ m): (a) the temporal waveform, in Fig. 7a; (b) the spectrogram, in Fig. 7b, as the scaled squared magnitude of short-time Fourier transform, in $Pa^2 \cdot Hz^{-1}$; and (c) the PSD, also expressed in $Pa^2 \cdot Hz^{-1}$, computed by either averaging or medianizing the PSD spectrogram, in Fig. 7c. The presented PSD curve shows two main bandlimited phenomena: (1) from 10 to 1000 Hz, which does not represent the bedload process but hydrodynamic processes; (2) from 1000 Hz to 50000 Hz which truly represents the bedload transport. The inversion procedure to estimate the GSD will be reliable as long as the bedload bandlimited region does not interfere with other extraneous noise source (hydrodynamic noise, turbulence). In the Isère River experiment, the SGN signal measured by drifts is almost free of hydrodynamic noise, which is proved by the typical median spectrum presented in Fig. 7c. In this study the inversion will be applied on such high signal-to-noise ratio PSD curves.

As in (Geay, 2013; Merchant et al., 2013), the median procedure is used to provide better smoothing as it better filters the unwanted low-frequency noises. In Fig. 7c, it can be noticed the suppression of lower frequencies spikes, attributed to hydrodynamic noise, when median PSD is used instead of the average one.

### 4.1.3    Pressure-difference sampling

A Toutle River (TR) sampler, depicted in Fig. 5b, has been used to sample bedload particles (entrance width of 305 mm by 152 mm). There were two mesh sizes used for sampling: 0.2 mm and 1.3 mm. Sample durations were between 4 and 8 minutes. Finally, each bedload sample was dried, weighted and sieved in the laboratory. The sampled sediments were classified into six size classes: $K = \{< 0.5; 0.5\text{-}2; 2\text{-}8; 8\text{-}16; 16\text{-}32; 32\text{-}64\}$ mm. The TR sampler has been deployed in three cross-sectional positions (at $X = 27$ m, $X = 35$ m and $X = 44$ m, marked on the bridge from left to right river banks). The number of repetition for each cross-sectional position is indicated in the Table 3. Bedload flux ($g \cdot s^{-1} \cdot m^{-1}$) have been averaged for each position of the sampler. GSDs have been computed for each position and for each mesh size used.

## 4.2    Results

### 4.2.1    Direct measurements of bedload

Results of TR sampler measurements are shown in the Fig. 8(a)-(b). A maximum of bedload flux was found in the middle of the cross-section ($X = 35$ m), Fig. 8a. A value of 100 $g \cdot s^{-1} \cdot m^{-1}$ has been measured. On side positions, the flux was found to be 5 times smaller, around 20 $g \cdot s^{-1} \cdot m^{-1}$. Concerning grain size distributions, most of the measurements indicate a $D_{50}$ between 7 and



20 mm. Notice that measurements made with the 0.2 mm mesh size towards the left bank ($X = 27$ m) indicate a GSD toward much finer sediments ($D_{50}$ of about 0.3 mm), Fig. 8b. Bedload samples closest to the left bank were indeed constituted of huge amount of fine sediment mixed with vegetable debris (about 60% of the total mass sampled). In the central and right positions, neither vegetable debris nor silts were sampled. TR sampler measurements showed grain size sorting along the river cross-section, varying from silts, near the left bank, to gravel, near the right bank.

In the following, the GSD measured in the central position ($X = 35$ m) will be considered. Its flux was indeed the largest measured and it is considered to be the principal source of bedload noise throughout the river.

### 4.2.2    SGN spectra inversion

All the median PSD of SGN signals recorded across the Isère River have been presented in Fig. 6b. The 7[th] drift will be studied, the one positioned in the centre of the cross-section at $X = 34$ m, which is the closest to the middle position of TR sampling measurements. Moreover, the most sediments were sampled in this position and the SPL map from Figure 7c showed that this position is the noisiest from all the cross-section. The results of spectrum inversion, using a modelled dictionary $\Delta$ with size classes from 1 mm to $K = 100$ mm (100 size classes), are shown in the Figure 9a. The results are compared to the GSD measured by the TR sampler near that position, more exactly in position $X = 35$ m. Four different values of the impact velocity U are tested (from 0.01 to 5 m·s$^{-1}$) and it is noticed that the impact velocity $U = 1$ m·s$^{-1}$ leads to a very good match between estimation and TR sampling measurements. This value of impact velocity will be used in the inversion of all other spectra measured across the Isère River.

Secondly, the GSD variations, represented by the percentiles $D_{16}$, $D_{50}$ and $D_{84}$, are estimated by the inversion of 12 drift measurements taken across the Isère River. Only the impact velocity of 1 m·s$^{-1}$ will be considered whereas the rest of parameters are defined in Table 1. The estimated percentiles are compared to equivalent diameters computed by the regression laws found in the paper (Thorne, 1986) defined below in Eq. (14) and Eq. (15), where the $D_{eq}$ are computed from the $f_{peak}$ and, respectively, $f_{centr}$. They are also compared to the TR sampler measurements, in the three positions across Isère River.

$$f_{peak} = \frac{224}{D_{eq}^{0.9}} \tag{14}$$

$$f_{centr} = \frac{209}{D_{eq}^{0.88}} \tag{15}$$

$$\int_{f_1}^{f_{centr}} P_{P\Sigma\Delta} df = \int_{f_{centr}}^{f_2} P_{P\Sigma\Delta} df \tag{16}$$

where $f_{centr}$ is centroid frequency, $P_{P\Sigma\Delta}$ is the PSD and $(f_1, f_2)$ is the frequency band defined by a certain value of power below the PSD's power peak.

It is observed that the estimated $D_{50}$ by NNLS algorithm is 1-2 mm matches very well to the $D_{50}$ measured by the TR sampler (circa 7 mm), in the middle of the river $X = 35$ m. On the other hand, the percentile $D_{16}$ matches the equivalent diameter



$D_{eq}$ estimated by Thorne's regression law $f_{centr}(D)$, Eq. (15), which is in average 50% below the measured $D_{50}$ by TR sampler. On the other hand, the percentile $D_{84}$ is close to the equivalent diameter $D_{eq}$ estimated by using the peak frequency regression law $f_{peak}(D)$, Eq. (14), overestimating the measurements of TR sampler.

## 5       Discussion on real data results

This work deals with development of a novel estimation strategy of bedload GSD from acoustic PSD. The spectrum inversion used the model based on sphere-slab impact, where the impacting sphere diameters range from $K = 1$ mm to $K = 100$ mm. The inversion of field experiments on the Isère River have shown in Figure 10a interesting results in conformity with the assumed hypothesis enounced in Sect. 2.4.

The inversion considered 4 values of impact velocity U = {$10^{-2}$; $10^{-1}$; 1; 5} m·s$^{-1}$. The best fit to the measured GSD by the
TR sampler, is when the impact velocity U tends to 1 m·s$^{-1}$ which could be possible for a large gravel bedded river like Isère. To verify this, the empirical model of (Sklar & Dietrich, 2004) was used to estimate the impact velocity for each grain size class. The model uses regression laws linking experimentally observed sediment velocities to the transport stage (the ratio between Shields shear stress and the critical shear stress). Using the Isère River's geometry as inputs: width = 60 m, depth = 2.5 m and slope = $5.5 \cdot 10^{-4}$ m·m$^{-1}$, the transport stage is computed and so impact velocity averaged over grain sizes impact
velocity is found to be of value of 0.9 m·s$^{-1}$. This estimated value is observed to be in accordance with the previous velocity observed from the inversion by NNLS algorithm.

The cross-sectional variation of the estimated $D_{16}$, $D_{50}$ and $D_{84}$ by the NNLS algorithm follows quite decently the trend of bedload $D_{50}$ measured by the TR sampler, Fig. 9b. By doing the acoustic inversion, it is noted the grain size sorting from the left to the right bank. However, it can be observed that the diameters monitored with the hydrophone are less variable along
the cross-section than in the case of TR sampler's measurements. This is explained by the fact that the hydrophone has the spatial integrative characteristic (Geay, 2017). The phenomenon of signal integration is typical for rivers like the Isère River, where high fluxes of bedload transport is concentrated only in a small portion across the section, i.e. in its centre. In this case the assumption of homogeneous spatial admitted in Section 2.4 is no longer valid. However, the powerful acoustic source makes noise all over the cross-section so it is like the sound sources are ubiquitous. This may be the reason that the inversion
of acoustic PSD measured in the centre ($X = 34$ m), for $U = 1$ ms$^{-1}$, still shows a good match to the sampling measurements in that position, only because of the high powerful acoustic source localized in this position.

Despite the consistent variation of the GSD across the river bed, measured by the sampler, the acoustic spectrum shapes shown in Fig. (6b) are relatively stable, in the interval $5 \cdot 10^{-4}$ and $5 \cdot 10^{-3}$ Pa·Hz$^{-1}$. This suggests that measurements by hydrophone installed from one of the banks are not dramatically different from measurements by free floating hydrophones along the
watercourse.



The propagation of sound throughout the local reach also raises some concerns about the quality of measured acoustic spectra. The model Eq. (9) is valid if the acoustic propagation only takes into consideration the sound divergence models. Thus, the acoustic propagation has effects on the spectral amplitudes but not on the spectrum's shape. In nature, however, there are many other acoustic propagation models in the river. One of them is when the high frequencies are more attenuated

than the lower ones. Also, higher frequencies are prone to scattering effects or to absorption, discussed in the context of river soundscape in (Tonolla et al., 2009). In the case of Isère Rivers, even though the suspended sediment transport is important, these effects are assumed to be mitigated by the fact the sound production from the powerful acoustic source from the centre overtakes assures enough good SNR of recorded signal. Another propagation effect concerns the lower frequencies, which are attenuated by the frequency cutoff phenomena, due to acoustic propagation in shallow waters, or waveguides (Geay, 2013;

Jensen et al., 2011; Rigby et al., 2016). During experimental fields, the Isère River has enough depth, 2.5 m, that the cutoff phenomenon cannot affect the generate frequencies associated to SGN. The pebble-sized particles that are up to 64 mm give SGN of dominating frequencies well above 1000 Hz, whereas the channel's depth of 2.5 m fixes the cutoff frequency to about 148 Hz, assuming a perfect rigid bottom. Therefore, the spectra in the bedload bandwidth will not be exposed to frequency cutoff so this does not present any risk to inversion. Yet, SGN monitoring and inversion technique for GSD determination is

particularly adapted to large rivers.

At first sight, our comparison with Thorne's regression laws would be very naïve due to the nature of theories: we considered the sphere-slab impact whereas the regression laws are from sphere-sphere impact phenomena. Therefore, the inversion is put into discussion when the bed river is no longer armoured and so, the model of impact between sphere and slab is debatable. Here, the target are the large gravel rivers. The dictionaries $\Delta$ for both impact models uses an impact velocity of

1 ms$^{-1}$, the material is granite and the log-normal distribution $\Gamma_{PMF}$ used in Eq. (9), has the shape $D_{84} = 2 \cdot D_{50}$, shown in Fig. 10a. It is noticed the shapes of simulated PSD, i.e. the two frequency peaks $f_{peak}$ and the spectra slopes, are found to be quite similar. Using the two dictionaries, the spectrum measured in the centre of river Isère ($X = 34$ m) is inversed. The Fig. 11b shows that the two solutions show no major disparities. This proves that sphere-slab framework modelling the collision between sediments and the bed river could work not only for exceptional hydraulic events but also for stable hydraulic

conditions.

Another strong hypothesis used in modelling PSD model of mixed impacts is that the particles are of spherical shapes. It is intuitively reasoned that the particle sphericity, shape factor and roundness also affect the acoustics of impacts. There are multiple possible ways of reckoning the equivalent diameter of a non-spherical particle. The particle's radius may be computed with respect the curvature of the region of contact, see (Chadwick et al., 2012; Goldsmith, 2003), with respect to the particle's

mass centroid (Thorne, 1986) or with respect to the $b$-axis of the particle (Wyss et al., 2016). Laboratory tests were conducted at GIPSA laboratory, during which two pebbles of size in the range 32 mm were impacted in a water pool along the three ellipsoid axis, $a$, $b$, $c$. The methodology of measuring the ellipsoid axis is found in (Bunte & Abt, 2001). It was found that the measured centroid frequencies takes values from 3000 to 8000 Hz. If regression law Eq. (14) is used, then the estimated diameters run from 23 mm to 73 mm which is happening to be the repartition of all possible radii of curvature of the respective





zones of contact. If the mode of sediment transport by sliding is the most frequent, then the particle *c*-axis could be used to infer an equivalent diameter. If the rolling mode is more frequent then the *b*-axis would be more appropriate to work with. Finally, if the saltation is concerned, which makes the point of this work, then axes *a* and *b* are equally probable to be taken into account in modelling impacts.

**6.      Conclusion**

A new strategy has been presented for data processing on hydrophone measurements for monitoring the bedload GSD in a gravel-bed river. This strategy defines a forward model and a spectrum inversion approach. Firstly, the forward model combines generated spectra from collisions between a sphere and a slab. Secondly, the inversion procedure treats the forward model as a linear system of equations and uses algebraic methods of solving least square problems to obtain the GSD.

The forward model is based on a weighted sum of analytical spectra developed in this paper and modelling the physics impact between a sphere and a slab. The weighting coefficients of the model represents a probability mass function which gives in the end the grain size distribution of bedload particles. The global sensitivity analysis on the PSD model of mixed impacts determined that the shape of GSD has the biggest influence on the shape of acoustic spectrum computed by Eq. (9). Another important parameters are the median diameter and the impact velocity. However, the influences are from mixed

interactions of parameters and it is very hard, if not impossible to obtain a complete analysis on the sensitivity of analytical model of Eq. (9).

     The PSD model of mixed impacts is working under the following strong assumptions: (1) the GSD is distributed everywhere in space and in the same way, (2) the acoustic propagation is not frequency-dependent and, so, the spectrum shape is not affected by propagation in river, (3) the impact velocity is invariant with the grain size, (4) the impacting particles are

20 of spherical shape.  The in situ experimentations showed that the integrative sound from all over the reach could render the first assumption verified (or true). In the case of the Isère River, the concentration of high transport rates in the middle of the cross-section permits reliable measurements of bedload GSD by hydrophone from river banks.

     The inversion method is a Non-Negative Least Square algorithm and it eliminates the negative solutions caused by ill-conditioned matrices. Concerning the least square approach for inversion, it is robust to noise.

The inversion of spectra from field trials on the Isère River proved that the method is highly reliable with no consideration of a priori information on bedform morphology of hydrological conditions. Surrogate methods for sediment transport in rivers were conceived in the idea of having access to information across all over the reach and real time. Contrary to geophones and Japanese pipe, the hydrophone    technique does not require special effort to be installed in the watercourse. As the method is one of the cheapest in terms of backing electronics, the hydrophone technique is still promising so further researches are to be

realized to develop it.





## Acknowledgement

The study was supported by funding of doctoral studies from the Rhône-Alpes Auvergne region through the "*Communautés de Recherche Académique*" (ARC N° 3) program for TP, by funding of research grant for TG from the convention n°C43R5T5030 between Électricité de France (EDF) and Grenoble Institute of Technology  and of research grant CHORUS for CG.

## Appendix A

### Table A1: Notations

| | | |
|---|---|---|
| $a, b, c$ | length of sediment (ellipses) axis | mm |
| $A$ | acceleration profile due to hertzian impact | m·s$^{-2}$ |
| $c$ | sound celerity in water | m·s$^{-1}$ |
| $\varDelta$ | modeled dictionary of individual spectra | |
| $d$ | power value of collision in a frequency narrow band | Pa$^2$·Hz$^{-1}$ |
| $E$ | elastic modulus (Young's modulus) of rigid body | Pa |
| $D$ | generic notation for the grain diameter | mm |
| $D_{eq}$ | equivalent diameter of the circle whose center is the grain's mass center | mm |
| $D_i$ | grain size belonging class $i$ from 1 mm to $K$ mm | mm |
| $D_{TR}$ | grain size class measured by Toutle River TR sampler | mm |
| $D_{16}, D_{50}, D_{84}$ | the 16$^{th}$, 50$^{th}$ and 84$^{th}$ percentiles of the grain size distribution | mm |
| $f$ | linear frequency | Hz |
| $f_{centr}$ | centroid frequency computed on the power spectrum density | Hz |
| $f_{peak}$ | peak frequency computed on the power spectrum density | Hz |
| $FAST$ | Fourier Amplitude Sensitivity Test | |
| $F_s$ | Sampling frequency | Hz |
| $\mathsf{F}$ | Fourier Transform operator | |
| $\mathsf{F}_{imp}$ | complex magnitude spectrum of the elastic impact, Eq. (5) | Pa·Hz$^{-1}$ |
| $|\mathsf{F}_{imp}|^2$ | noise power spectrum of the elastic impact, Eq. (6) | Pa$^2$·Hz$^{-1}$ |
| $GSA$ | Global Sensitivity Analysis | |
| $GSD$ | grain size distribution | |
| $\varGamma_{PMF}$ | solution of inversion in the form of probability mass function (the set of $n_i$) | |
| $\varGamma_{GSD}$ | solution of inversion in the form of grain size distribution | |



| $j$ | imaginary unit | |
|---|---|---|
| $K$ | number of grain sizes classes | |
| $LS$ | least square problem | |
| $v$ | Poisson's ratio of rigid body | |
| $n_i$ | the weighting coefficient of the spectrum produced by the size class $i$ sphere | |
| $N_{FFT}$ | number of narrowband power values composing the power spectrum density | |
| $N_{sig}$ | number of samples in the impact signal | |
| $NNLS$ | non-negative least square problem | |
| $\omega$ | angular frequency | rad·s$^{-1}$ |
| $P(\omega)$ | power spectrum density of the noise from an elastic impact, Eq. (7) | Pa$^2$·Hz$^{-1}$ |
| $PMF$ | probability mass function | |
| $P_{P\Sigma\Delta}$ | power spectrum density of the sediment size mixture, Eq. (9) | Pa$^2$·Hz$^{-1}$ |
| $PSD$ | power spectrum density | Pa$^2$·Hz$^{-1}$ |
| $r$ | reference measurement distance between the sensor and the center of the impact (see Fig. 1a-b) | m |
| $\rho_s$ | density of sediment | kg·m$^{-3}$ |
| $\rho$ | density of water | kg·m$^{-3}$ |
| $SGN$ | self-generated noise (noise generated by sediment collisions) | |
| $S_i$ | sensitivity indices from the first order global sensitivity analysis | |
| $\sigma$ | the standard deviation of a normal distribution (used in sensitivity analysis) | |
| $T_d$ | phase shift between the two signals from the recorded impact | s |
| $t$ | time | s |
| $t'$ | delayed time | s |
| $\theta$ | angle of directivity acoustic sources – sensor | ° |
| $T_c$ | duration of hertzian contact | s |
| $T_d$ | delayed time (delayed propagation due to the geometry of particles) | s |
| $U$ | impact velocity | m·s$^{-1}$ |
| $X$ | position on the cross-section of the Isère River (marked on the bridge from left to right bank) | m |



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



**Table 1: Parameters used to model analytical spectra of sediment size mixtures, Eq. (9), and the typical values adapted for underwater environment. The typical singular values are used in inversion further in this paper. The ranges of values are used in the global sensitivity analysis. $K$ is the number of size classes used to inverse acoustic spectra.**

| Parameters | Typical range of values in underwater medium | Typical values used in inversion | Units | Remarks |
|---|---|---|---|---|
| Particle diameter ($D$) | 0-150 | {1,2, …, 150} | mm | The mean $\mu = \ln(D) = D_{50}$ is used in the global sensitivity analysis (GSA) |
| Standard deviation ($\sigma$) | 0.01-10 | $2D_{50}=D_{84}$ | mm | Used in the GSA; the relation $2D_{50}=D_{84}$ typical refers to surface GSD |
| Impact velocity ($U$) | 0.001 … 5 | {0.01; 0.1; 1; 5} | ms$^{-1}$ | The same for all the grain size classes |
| Distance of measurement ($r$) | 0.01 … 10 | 1 | m | It acts on the delay time $T_d$ found in the model of Eq. (7) |
| Angle of directivity ($\theta$) | 0°…90° | 0° | deg | In theory, if $\theta = 90°$ then the wave amplitude is zero; it also acts on $T_d$ |
| Sound celerity in water ($c$) | 1403-1507 | 1483 | ms$^{-1}$ | Dependent on temperature, water salinity, etc. |
| Water density ($\rho$) | 960-1025 | 999 | kgm$^{-3}$ | Dependent on temperature, water salinity, etc. |
| Modulus of elasticity ($E$) | 10-70 | 55 | GPa | Materials like limestone, quartz, granite. The typical values are for granite. |
| Poisson's ratio of impacting bodies ($v$) | 0.15-0.2 | 0.2 | - | |
| Density of sphere ($\rho_s$) | 1800-2750 | 2700 | kgm$^{-3}$ | |





**Table 2: First order sensitivity indices $S_i$ computed by the FAST method, assuming the peak frequency as the output of the model, $f_{peak}$: 0% means no influence, 100% means total influence on the model output**

| Input parameters | First order sensitivity indices $S_i$ % |
|---|---|
| $\sigma$ | 41.96 |
| $D_{50}$ | 21.29 |
| $U$ | 12.71 |
| $E$ | 8.87 |
| $r$ | 5.98 |
| $v$ | 5.52 |
| $\rho_s$ | 1.75 |
| $\theta$ | 0.97 |
| $c$ | 0.62 |
| $\rho$ | 0.34 |

5 **Table 3: Number of repetition for each measurement**

| Position on cross-section $X$ (m) | Mesh size of 0.2 mm | Mesh size of 1.3 mm |
|---|---|---|
| 27 | 3 | 3 |
| 35 | 2 | 2 |
| 44 | 1 | 2 |



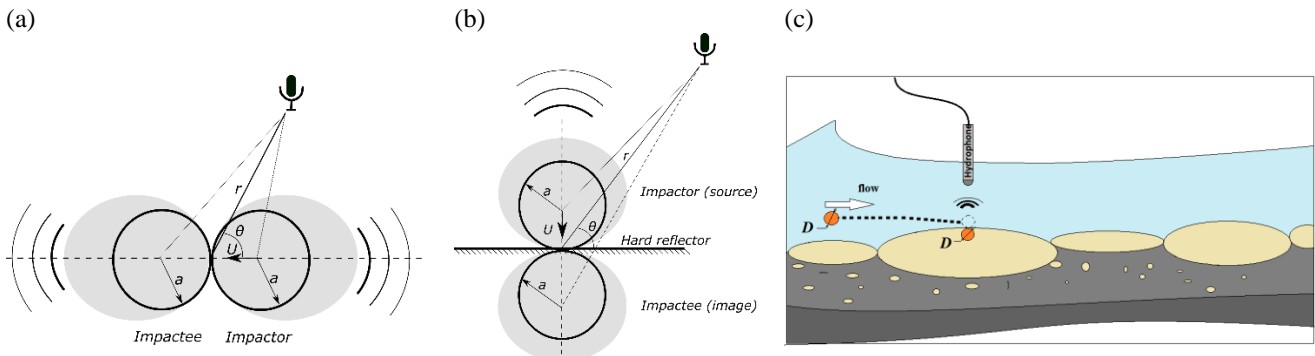

**Figure 1 : (a) Setup for the impact between two spheres, here of same radius $a$, where the two wavefields are summed up; the dipole acoustic source is illustratively depicted by the gray patch; (b) Setup for the impact between a sphere of radius $a$ and an semi-infinite rigid plane; to be noted the boundary condition of hard bottom (reflector) assumed in order to use the 'method of images'; consequently, in the acoustic treatment, the impacting sphere is mirrored in the slab and the wavefields are subtracted; the dipole acoustic source is illustratively depicted by the gray patch; (c) the elementary acoustic process of bedload noise in the river: the particle of equivalent diameter $D=2a$ impacts the armored bed river (a massive slab) which generates a transient measured by hydrophone;**





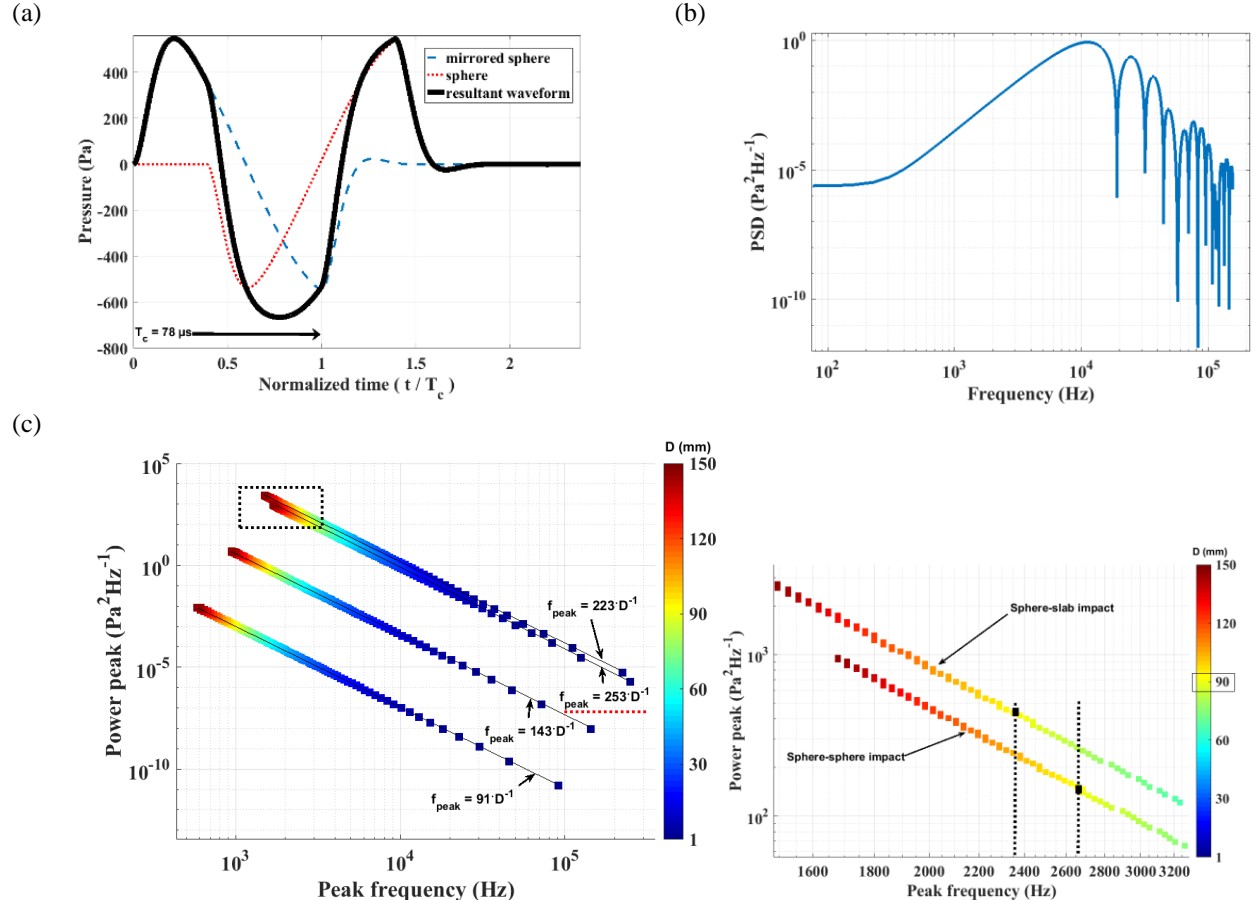

**Figure 2 : a) Analytical waveform of sound from impact between a granite sphere of radius $a$ = 10 mm and a granite slab, where the impact velocity $U$ = 1 m·s$^{-1}$, the directivity angle $\theta$ = 0° and the sensor is at $r$ = 1 m from the impact; the arrow indicates the contact duration $T_d$; b) The analytical spectrum modeled with Eq. (7) using the same parameters as in figure (a); (c) peak frequency $f_{peak}$ and power peak variations, from spectra modeled by Eq. (7), with diameter and sphere's diameters; the diameters are coded by colors,. The power law $f_{peak} = aD^b$ is given, where the sphere-slab impact tests consider three impact velocities ($U$ = {0.01, 0.1, 1} ms$^{-1}$) and the law of sphere-sphere impact is underlined by dotted line; the material is granite. From bottom to top, the regression laws of sphere-slab impact vary from $U$ = 0.01 (bottom) to $U$ = 1 (top) ms$^{-1}$. The sphere-sphere impact tests are made using $U$ = 1ms$^{-1}$ and the same other parameters as sphere-slab impacts. In the detail, at right, the two vertical dotted lines locate the $f_{peak}$ of impact spectrum from 90 mm –diameter particles for both impact models.**




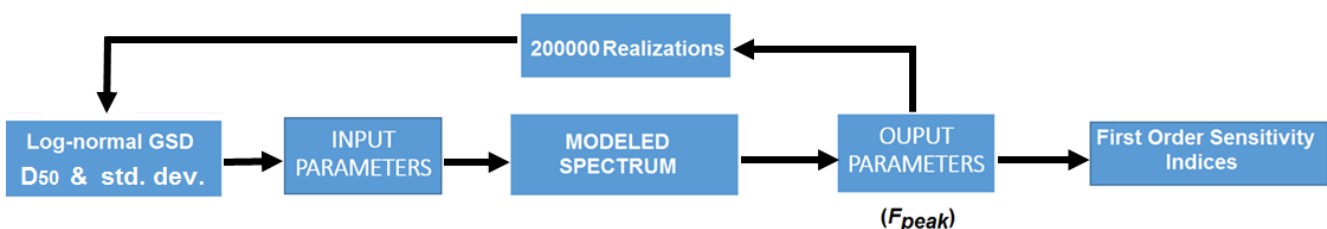

**Figure 3: GSA flowchart to compute the first order sensitivity indices by *FAST* method; the spectrum is simulated with Eq. (9), from a lognormal distribution generated using the mean diameter and standard deviation $\sigma$ associated to a normal distribution. The associated mean of the normal distribution is equal to log($D_{50}$), where $D_{50}$ is the median diameter of the log-normal distribution. The rest of input parameters used in the analytical model from Eq. (7) are defined in Table 1. From the simulated spectra, the $f_{peak}$ are computed and finally the first order sensitivity indices $S_i$ are calculated. The results are shown in Table 2.**





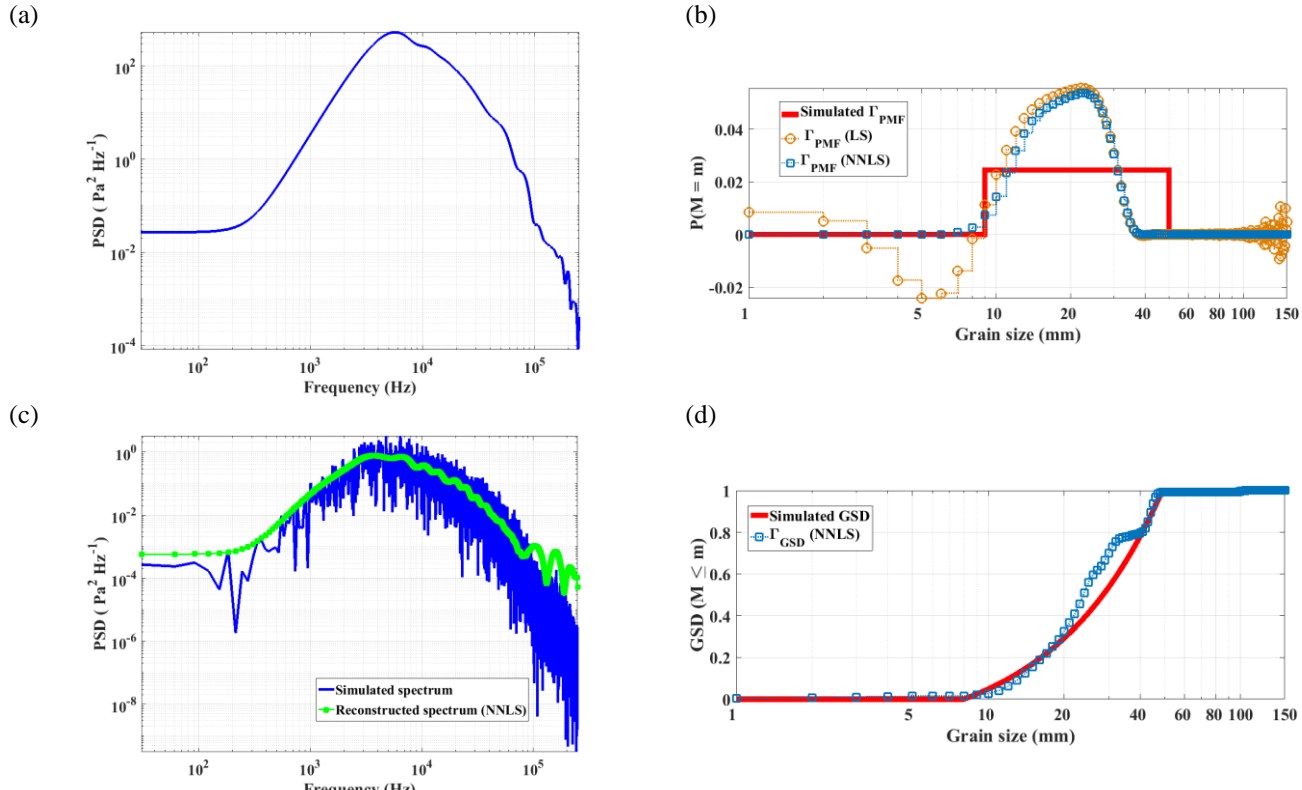

**Figure 4: (a) Simulated** *PSD* **from the uniform** *PMF* **of mass of sediments, 10 kg per 1 mm-size class, from 10 to 50 mm, where the impact velocity is** $U = 1$ **ms$^{-1}$; the other input parameters are defined in Table 1; (b) the PMF solutions obtained from the inversion of spectrum shown in figure (a) using the two algebraic methods: the classical LS with pseudoinverse and the NNLS algorithm. The impact velocity used in modeling the dictionaries of impact spectra was** $U = 0.1$ **ms$^{-1}$ (to note the difference between this one and the one used in simulation,** $U = 1$ **ms$^{-1}$) and the other input parameters being the same as in simulation; (c) the simulated** *PSD* **from figure (a) but simulated for** $U = 0.1$ **ms$^{-1}$ and with variance added (see text for noise simulation procedure); (d) the cumulative GSD obtained from the inversion of the noised spectrum from figure (c), where for modelling the dictionary of spectra** $U = 0.1$ **ms$^{-1}$. The estimated GSD is used to reconstruct the spectrum, shown in figure (c).**

(a) 

(b)

(c)

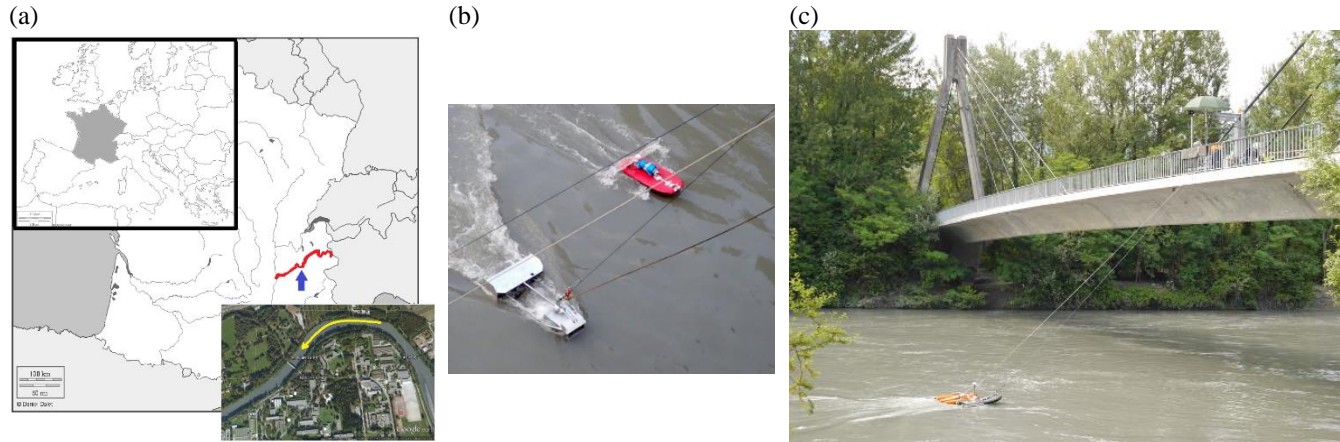

**Figure 5: Experimental setup: (a) Isère River basin geographical location (http://histgeo.ac-aix- marseille.fr) and Goggle Earth © picture showing the river morphology near the bridge where measurements were taken; (b) Instruments used during the trials; from left to right: Toutle TR sampler and the floating river-board with hydrophone; (c) the bridge from where acoustic transects and sediment sampling were realized.**



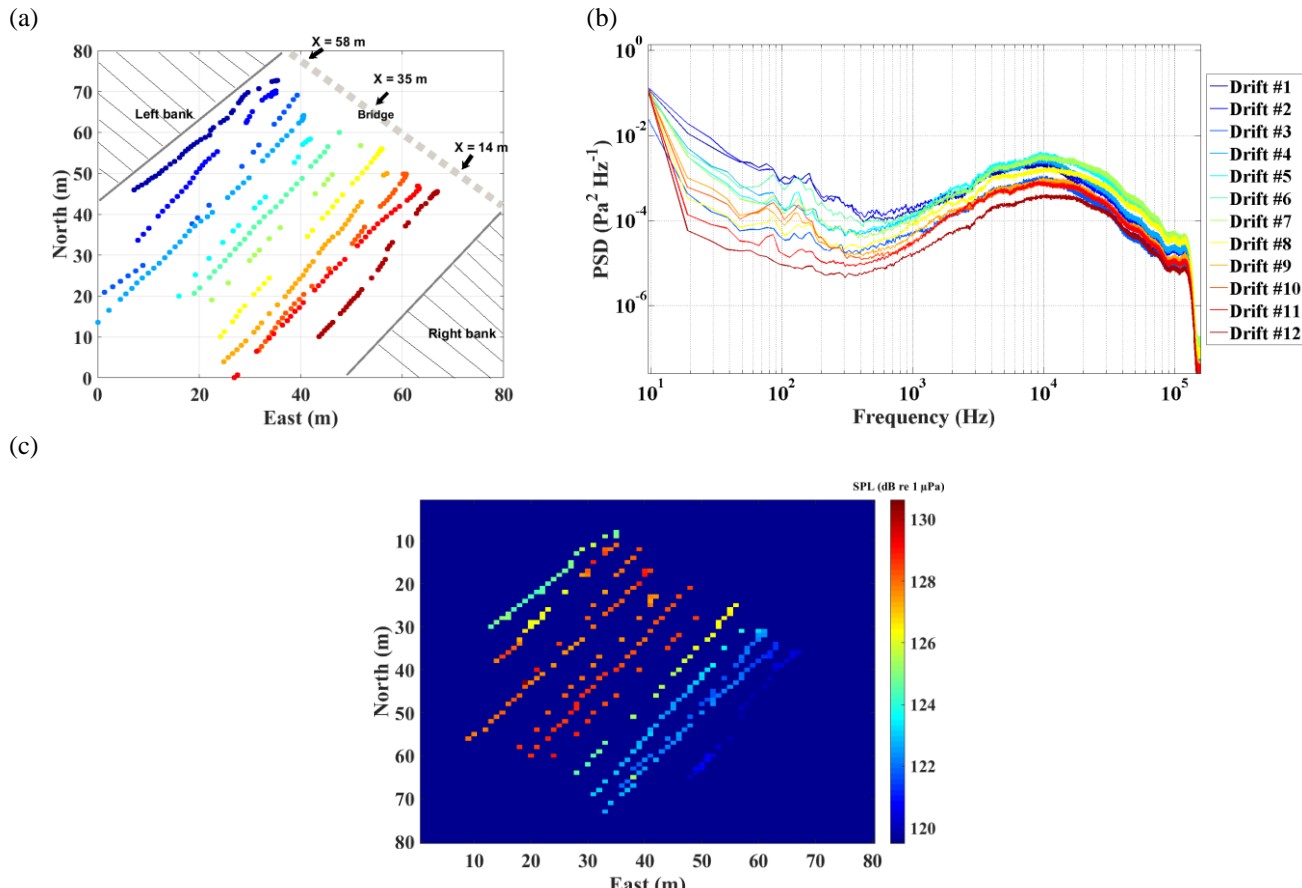

**Figure 6: (a) Positions of the floating board during drift measurements, with essential positions marked on the bridge,** $X = \{14, 35, 58\}$ **m across the river; (b) the** *PSD* **estimated from the 12 drifts, in units of** $Pa^2 \cdot Hz^{-1}$**; to be noted the change in peak frequencies: the leftmost position (Drift #12) has the highest frequency, meaning that the finer size fractions are transported, and the particles are getting coarser up to the right bank; (c) the measured** *SPL* **map from the 12 drifts, in in units of dB re 1 μPa; the maximum values are seen to be in the middle of Isère River's cross-section.**





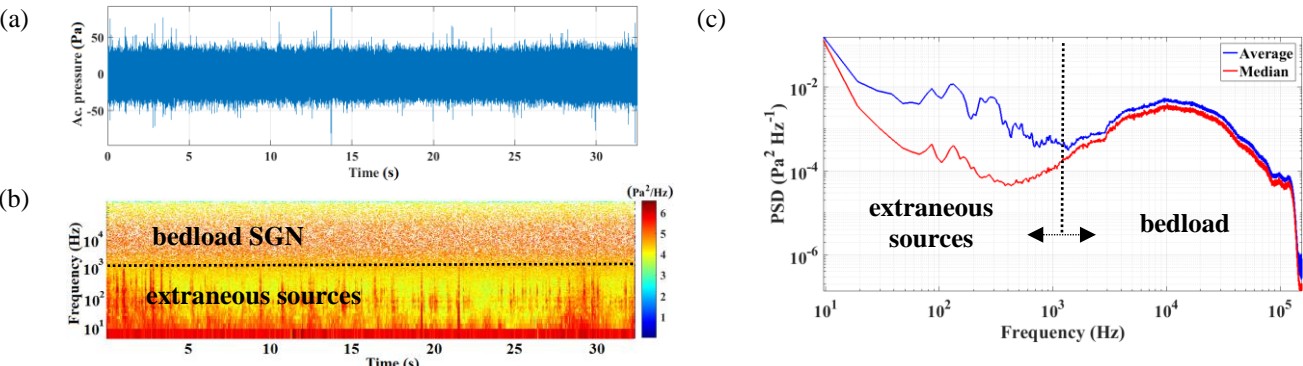

**Figure 7: Signal representations of the *SGN* recorded during hydrophone experiments on the Isère River (France): (a) temporal signal in units of Pa; (b) time-frequency representation (spectrogram), with the color code normalized with respect to power values, in Pa². Hz⁻¹; the specific frequency bandwidth of the bedload acoustic effects and of the hydrodynamic noise agitation (extraneous sources) are indicated ; (c) the *PSD* curve, also in Pa². Hz⁻¹, estimated using either the average or the median power values, in time, from the spectrogram in (b).**





(a)

(b)

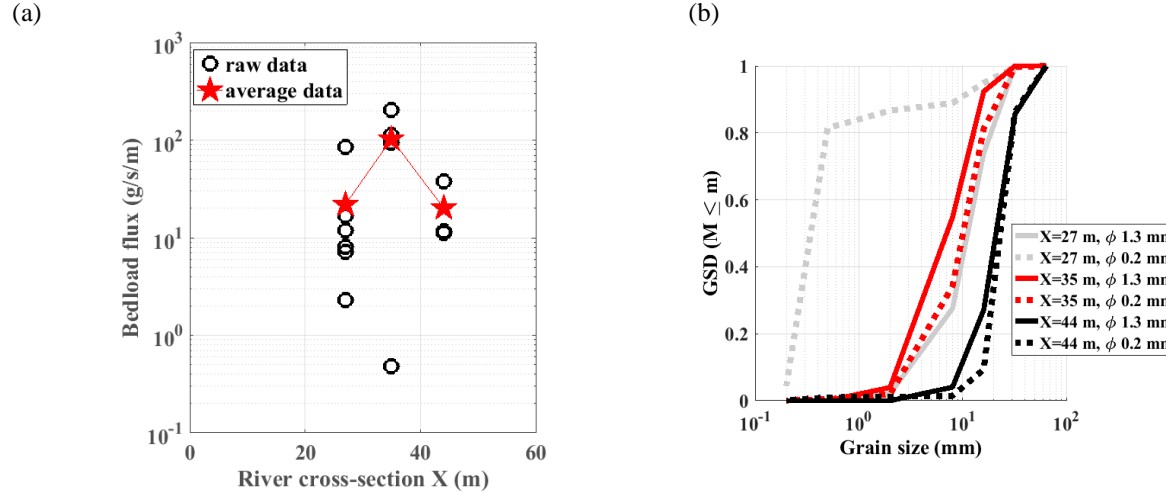

**Figure 8:** Measured bedload flux in three positions across the Isère River, $X$ = {27, 35, 44} m and (b) measured $GSD$ curves in these positions, using the TR sampler with two mesh sizes, 0.2 mm and 1.3 mm.





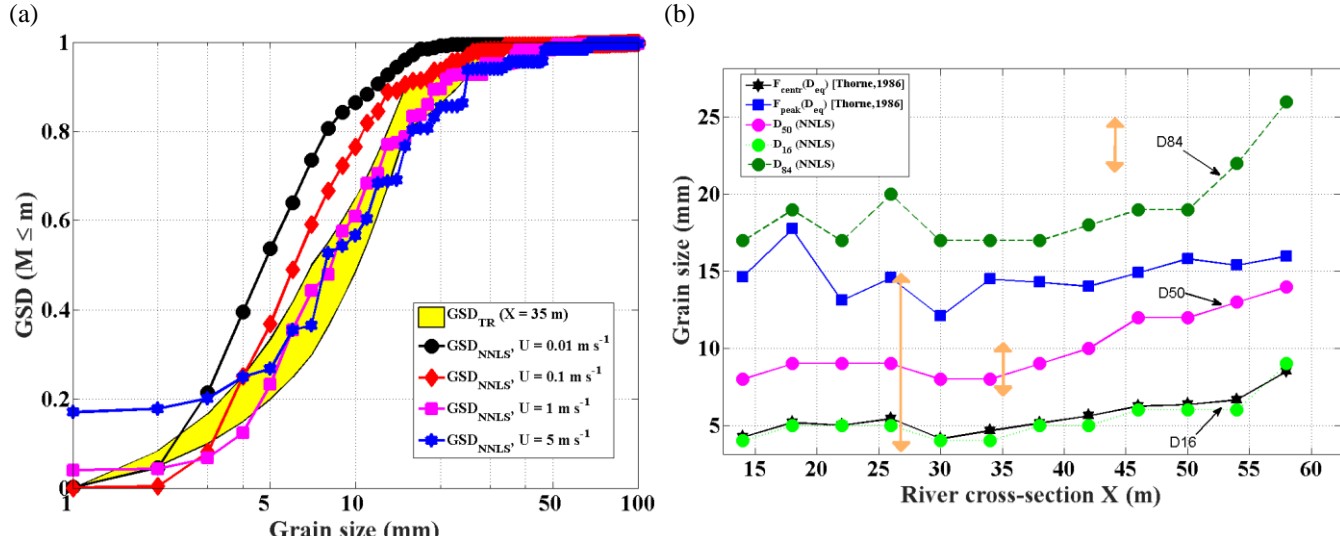

**Figure 9: (a) Estimated** *GSD* **by** *NNLS* **algorithm in the center of the Isère River** ($X = 34$ m)**, using different values of impact velocities** $U = \{0.01, 0.1, 1, 5\}$ m·s$^{-1}$**. Measured** *GSD* **by** *TR* **sampler** ($X = 35$ m) **is represented by the yellow envelope for the two mesh sizes (see Figure 9b for fraction sizes finer than 1 mm); (b) The** $D_{16}, D_{50}, D_{84}$ **estimated by** *NNLS* **across the Isère River, compared to Thorne's regression laws to estimate equivalent diameter** $D_{eq}$ **and to the range of** $D_{50}$ **measured by TR sampler (** $X = \{27, 35, 44\}$ m **), the vertical double arrows. The impact velocity used in inversion is** $U = 1$ ms$^{-1}$**.**





(a)

(b)

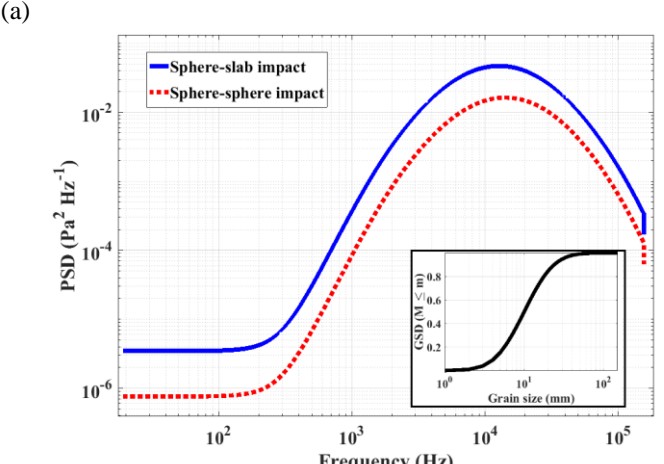

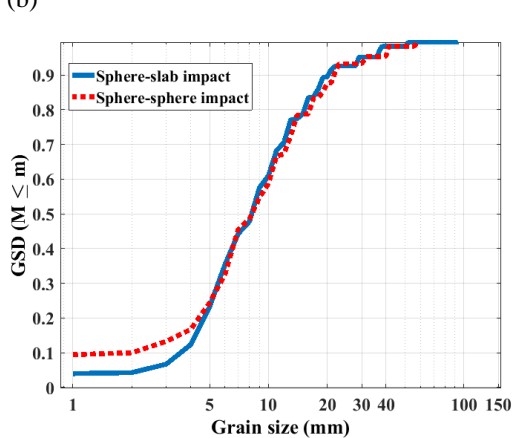

**Figure 10: (a) Modeled spectra using a log-normal *GSD*, D = 1, 2, … , 150 mm where $D_{84} = 2 \cdot D_{50}$, $D_{50} = 10$ mm (see medallion); typical input parameters are given in the Table 1 and $U = 1$ m s⁻¹. Concerning the sphere-sphere impact, the impactor has the same size as the impactee; (b) Inversion by NNLS algorithm of the acoustic spectrum measured in the center of Isère River ($X = 35$ m) using dictionaries of analytical spectra of sphere-slab and sphere-sphere impact, with input parameters defined in Table 1 and $U = 1$ m s⁻¹**