# Peer review of "Passive Acoustic Measurement of Bedload Grain Size Distribution using the Self-Generated Noise"

_Hydrology and Earth System Sciences, 2017_

## Referee Comment (RC1) · Anonymous Referee #1 · 14 Jun 2017

The manuscript deals with the extraction of the bedload particle mass distribution by monitoring the sound field in a fast flowing river. The manuscript is interesting, however, the matrix methodology used to obtain the distribution requires further explanation. As the text stands it is difficult for the reader to ascertain exactly how the mass distribution is being estimated in terms of physical processes due to the rather unclear meaning of the matrix formulation.

1. P2 Line 3 Parker is cited without a reference.

2. P4 Line 7 'and so integral' should be 'and so the integral'.

3. The authors need to provide more justification as to why they consider the sphere-

slab impact model is an improvement on the sphere–sphere impact model. The slab model uses an image of the impactor as the impactee and hence all collisions are between particles of the same size. The bed is not a slab of material, but a heterogeneous mixture of individual gravel particles of varying differing diameter and hence impacts between particles of differing diameter is highly probable.

4. P6 line 25 'to reduce eliminate' should be 'to reduce'.

5. P7 line 7. Is the term 'dictionary $\Delta$' a technical term, it is not one commonly used in matrix descriptions. What is being referred to by 'dictionary $\Delta$'?

6. A clearer description of the operation of equation 9 is needed, e.g. 'or put in another words' P7 lines 8/9, is not sufficient. This equation is the kernel of the manuscript and needs a high degree of clarity to help the reader understand the manuscript, particularly those who are unfamiliar with the matrix formulation and inversion. Clearly explain the contents of the rows and columns, the matrix operation and how the formulation relates to obtaining the GSD with equation (12). At present the text is loaded with matrix jargon, which makes it difficult for the non-expert in matrix manipulation to follow.

7. P8 line 10 It is unclear what is meant by ''but rather decent idea could be concluded on the model's behaviour', this needs re-phrasing.

8. P8 line 19, 'rest of the parameters are considered of little influence'. It would be helpful to the reader to specify which parameters.

9. P9 line 4 'thanks to' is somewhat colloquial, 'using the' may be more appropriate.

10. P9 The description of equation (12) is quite terse and for reader not familiar with matrix inversion difficult to understand. As with point 6 above, spending a little more time explaining the operation on the matrix formulation and inversion, with a description of the physics which is taking place, would make the analysis much more accessible. Matrix formulations and inversions though relatively common is still a very specialised area and require explanation.

11. P11 Line 26. Again the word 'thanks' is used when 'due to' would be more appropriate.

12. P12. In section 4.1.2 it is stated that 10-1000 kHz does not represent SGN and 1000-50000 kHz is SGN. Some justification is required for this statement.

13. P12 line 13 'almost free of hydrodynamic noise' and line 15/16 'attributed to hydrodynamic noise' seem slightly contradictory statements about the recorded SGN.

14. P13. 'D50 by NNLS algorithm is 1-2 mm', this is not a consistent result with figure 9 (a & b) which have a D50 of 8-18 mm.

15. P14 Line 2 'The model Eq. (9) is valid if the acoustic propagation only takes into consideration the sound divergence model.' It is not clear what the 'sound divergence model is', an explanation is required.

16. P15 line 8 'overtakes assures enough good SNR of recorded signal'. This is somewhat gobbledegook, what is actually meant.

17. P15 Line 24 the use of the word 'repartition' is unclear.

The authors use size, radius and diameter, to describe the dimensions of the spheres, size is ambiguous and should not be used as it could be either diameter or radius.

Before publication, further clarity on the matrix formulation and inversion is required, specifying clearly in physical terms what the columns and rows are in equation (9) and how in practice, i.e. the physical process in equation (13), which generate the GSD. Is it some optimum fit between the measured and computed PSD? It is not exactly clear what physical criterion is used in the matrix inversion to obtain the GSD in figure 9.

---

## Referee Comment (RC2) · Anonymous Referee #2 · 9 Oct 2017

This manuscript presents an inversion strategy of hydroacoustic data in order to estimate the grain size distribution of bedload material. The authors use a quite innovative approach in this research field, namely the monitoring of bedload transport using surrogate techniques, leading to interesting results. The applicability of the presented method is constrained by strong assumptions but this study remains of significant importance for the development of quantitative bedload monitoring techniques using acoustic sensors. The methodology is sound but the general presentation of the paper should be improved. Overall, I do not have any critical comments, most of them concern minor (typo) errors or a few points that need to be further clarified.

[Figure]

1. P2/L3: Reference (Parker, 1990) must be mentioned here.

2. P2/L5: If it concerns many scientists, it would be preferable to cite more than one article (or indicate "and references therein" if it is a review article).

3. P2/L7-12: Here you should cite appropriate references.

4. P2/L12: You could introduce here the term SGN, for instance at the end of the sentence ", so-called the bedload Self-Generated Noise (SGN)."

5. P2/L18: replace bear by contain.

6. P2/L22: "Geay et al., 2017" instead of "Geay, 2017".

7. P2/L24: The year for reference Krein et al. is wrong (2016 instead of 2014). You can also cite Wyss et al. (2016).

8. P2/L25: You should avoid the use of too many acronyms, notably here for a section title. Replace "SGN model" by "Bedload Self-Generated acoustic Noise (SGN) model" or something equivalent.

9. P4/L2-4: three times "mathematically" in three consecutive sentences. Please rephrase.

10. P4/L16: "The material parameter..." there is no verb in this sentence. Explaining equations and parameters can be done in a concise manner but, as a general comment for this theoretical part, your presentation sometimes suffers from a lack of clarity.

11. P4/L30: The acronym FT for Fourier Transform is not explained.

12. P5/L1-2: "is" is missing two times before "FT of hertzian..." and "angular frequency". The definition of the scalar, angular frequency should be expressed here.

13. P6/L1: It is good to provide an example of waveform computed with Akay's model but, for the sake of completeness, you could also provide the expression of Eq.7 from Akay and Hodgson (1978) as appendix.

14. P6/L25: You have to choose between "to reduce" or "to eliminate". However, this sentence appears also insufficient to justify the choice of the sphere-slab model. Could you make a link to the last section and figure ?

15. P7/L14: The reference to Eq.10a in this sentence is somewhat unclear. Please consider rewriting.

16. P8/L7: "ln(D) coincide the D50 is . . ." ?

17. P8/L18: "as it the phase shift" ?

18. P8/L23: three times the word "analysis" in the same sentence. Please rephrase.

19. P8/L28: "material of sediments" could be replaced by "material properties (i.e., density, Poisson's ratio, Young modulus)".

20. P8/L28: This is not 12% actually but a little bit more.

21. P9/Section 2.4: You should discuss some of these assumptions in light of the recent work by Geay et al. (2017, "Spectral variations of underwater river sounds", EPSL).

22. P9/L19-22: You should recall the meaning of acronym LS for least square at line 19 instead of line 22.

23. P9/L20-21 & section 3.1 L12-18: Refer to Figure 2c or, eventually, you could make a figure depicting several PSD for different size classes in order to easily gauge the variability of the acoustic spectra.

24. P9/L23: Consider replacing "solution" by "least square solution" or "minimum solution".

25. P10/L27-29: This part, which concerns the signal processing, is too short. You should give more details on the processing parameters you used.

26. P12/L2: "SGN map" should be replaced by "SPL map" as indicated in the figure.

27. P12/L8-10: You should briefly argument how you infer the probable dominant noise sources in their respective frequency bands using simulation results and previous studies.

28. P12/section 4.1.2: Did you consider testing smoothing functions over logarithmically spaced frequencies ?

29. P13/L11: Figure 6c instead of Figure 7c.

30. P13/L12: The term "noisiest" could be misleading since it can also refer to unwanted noise sources. You can add something like "in the bedload frequency band".

31. P13/L21: Use "by" instead of "in the paper". The notation Deq must be written next to "equivalent diameter "at line 20.

32. P13/L26: "a certain value", which one ?

33. P13/L28: "is 1-2 mm matches very well" ?

34. P14/L15: Please remove "observed to be".

35. P14/L17: "follows quite decently", not for X∼45m. You should already point out this discrepancy here.

36. P15/L2: Please consider rephrasing, for instance "The model in Eq.(9) only takes into consideration signal attenuation due to sound divergence, which means the acoustic propagation . . .".

37. P15/L5-15: Referring to Geay et al. (2017, EPSL) would be appropriate. In conclusion to this paper, this discussion part is useful to assess the advantages and limitations of the presented approach.

38. The conclusion (as well as the abstract) are generally written in a comprehensive way. The last paragraph in the conclusion could be eventually reformulated. Even though relevant, the cost of the technique is maybe not the best argument to point out

in the last sentence.

39. Tables and Figures: Overall, tables and figures support well the text. Event though not critical, legend and axis labels are too small and the arrangement of subfigures is not well balanced for most of the figures. Fig. 6c- Please consider to set the background color as white, as done for Fig. 6a. You could also report the cross-section on this panel and keep the same y-axis direction between Fig. 6a and 6c. Two "in" in the caption ("in in units of dB. . ."). Fig. 10- replace "medallion" by "inset". The title of table 3 is not enough explicit.

---

## Author Comment (AC1) · 30 Oct 2017

NOTE: The authors did a change in the Eq. (7) which will slightly changes the results of numerical tests and field spectra inversion presented in this article. The change has the physical meaning of passing from power to energy representation of impacts and so it avoids the usage of time in the impact modeling. As the clarity of the presentation was invoked by the two referees, we also did some changes regarding the system of notation in order to define easier-to-read equations. In consequence, some parts of the text change but the modifications still follow the pertinent advices of the referees. The resulted paper is also added in the supplement of this response comment. Below,

[Figure]

we are considering the referee's suggestions and corrections step by step.

1. Added reference (Parker, 1990) instead of 'Parker'

2. Correction applied.

3. On P6 Line 25 was given the argument for the use of slab-sphere impact instead of sphere-sphere impact. This means that the slab do not contribute to hertzian sound production as it does not oscillate; his role in the sound production is the reflection of the sound from the impactor particles, by which it allows the method of image modelling. As the impactee particles in the bed river are fixed, we may consider the bed river as a massive slab which reflects the hertzian sound pressure generated by impactor particles. In this way, we skip the task of determining the dimensions of impactee particles."

Thus, we rephrase:

"This could justify the reason to use the sphere-slab impact physics instead of sphere-sphere impact, to reduce eliminate the need for the dimensions of the impactee object"

to the following:

"In this paper, we choose to use a slab model to model bedload SGN as it simplifies the inverse problem. Indeed, the task of determining the dimensions of impacted particles is skipped. Therefore, we consider that the riverbed could be modeled as a slab. This hypothesis could be supported when the riverbed is armored or paved, but may be false when the river bed is totally mobile and when the impacts between particles of different diameters are very common."

4. Correction applied. Thanks!

5. The notation in this case does not have technical significance; we could have call it a 'catalog'. There is no work to approach this subject so we must invent a notation system for our spectrum model. We chose the Greek letter for 'D', from 'Dictionary',

and to not be confused with 'Diameter'. The dictionary contains analytical spectra for each 1-mm size class, from 1 mm to 150 mm. Thus, the dictionary is a matrix whose columns are spanned by analytical spectra. The explanation is made after Eq. (9a-b).

6. Considering the present system of notation we rephrase the paragraph including the P7 lines 8/9 :

"where PP$\Sigma\Delta$ is the PSD of the bedload, $\Gamma$PMF is a probability mass function (PMF) of the number of collisions, and Pi is the analytical elementary PSD of the impact between spheres of the size class i ( Eq. (7)). The matrix $\Delta$ contains the analytical spectra of impacts for each size classes . The size class i takes integer values, from the lowest limit, 1 mm, to the highest one, K mm, where K is the largest diameter considered in modelling. Here, the studies considers K equal to 150 mm. Thus, the ni represents the probability to observe an impact of the particle of the size class i."

to the following:

"where $\Delta$ is the dictionary of elementary ESD of impacts between spheres and slab and I is the vector of impact rates per diameter class or, basically, a histogram. The class i takes integer values, from the lowest limit, 1 mm, to the highest one, K mm, where K is the largest diameter considered in modelling. Here, the studies considers K equal to 150 mm. The parameter NFFT is the number of values contained in the spectrum or the number of Fourier Transform points on which the spectrum is modelled."

7. Rephrasing the line to the following:

"but pertinent ideas could be drawn on the model's behaviour"

8. Rephrasing the line to the following:

"The angle of the point of observation with respect to the impact, $\theta$, and the propagation medium properties,  and c, are considered of little influence on the values of fpeak".

9. Replacement applied. Thanks!

10. Equation redefined according to new system of notations and without the argmin operation and least square framework.

11. Replacement applied. Thanks!

12. To explain this, the following statement is added on P12 Line 8:

"Two main sources of noise can be distinguished in the recordings: below and above 400 Hz (fig. 7). Bedload impacts can clearly be heard in the higher frequency band, it sounds like the crackling of the flames. Sounds occurring below 400 Hz are not propagating sounds as they are localized below the cutoff frequency of the river waveguide (Geay et al., 2017b; Rigby et al., 2016).".

13. Indeed, low frequency noises have two origins. The continuous noise below 400 Hz is probably related to turbulence induced noise around the sensor. A second type of noise (impulsive) can be observed in the spectrogram. This noise is due to some mechanical movements of the structure sharing the hydrophones. The median filter enables to filters these unwanted/intermittent noises.

Added remark on P12 Line 13: "The median procedure is used to provide better smoothing as it better filters the unwanted low-frequency noises (Geay et al., 2017a). "

14. Thanks, it has been changed. Correction: "1-2 mm" to "10-14 mm".

15. Rephrasing:

"The model Eq. (9) is valid if the acoustic propagation only takes into consideration the sound divergence models."

to the following:

"The proposed model (eq. 9b) has been elaborated by assuming a simple geometrical spreading model of the acoustic waves in the river."

16. The word "overtakes" is removed from the statement. Thanks!

17. In P15 Line 30, the word "repartition" is replaced with "range". The statement is reformulated.

Rephrasing:

"If regression law Eq. (14) is used, then the estimated diameters run from 23 mm to 73 mm which is happening to be the repartition of all possible radii of curvature of the respective zones of contact."

to the following:

"If regression law Eq. (14) is used, then the estimated diameters span the range from 23 mm to 73 mm which is the repartition of all possible radii of curvature of the respective zones of contact. "

18. "Size" is linked to particles or grains which is equal to an equivalent diameter, Deq, because the real grains are not perfectly spherically. On the other hand, the terms "radius" and "diameter" are attributed to spheres. Parameter "radius" is also present in the equations of hertzian impact for sphere impacts.

19. Following the recommendations in the last point, we decided to do changes in the system of notations to clarify as much as possible the formulation. The modified version according to all of the above points is in the supplement of this comment.

Please also note the supplement to this comment:
https://www.hydrol-earth-syst-sci-discuss.net/hess-2017-171/hess-2017-171-AC1-supplement.pdf

**Supplement:**

[revised manuscript text omitted]

$$\mathsf{F}_{acc}(\omega) = \mathsf{F}(p_I(t)) \cdot \mathsf{F}(A(t)) \tag{4a}$$

where

$\mathsf{F}(p_I)$– is the Fourier Transform (FT) of Kirchhoff's impulse response $p_I$ (Koss and Alfredson, 1973), for a sphere of radius $a$, defined in the Eq. (4b),

$\mathsf{F}(A)$ –the FT of hertzian acceleration due to elastic impact between two same radius and same material spheres, defined in Eq. (4c), and $\omega$ is the angular frequency which is a measure of rotation rate, in radians per seconds, and it is equal to $2\pi f$, $f$ is the linear frequency, a measure of number of occurrences per second.

$$\mathsf{F}(p_I) = \frac{\rho a^3 c}{r^2}(c + j\omega r)\frac{2c^2 - (\omega a)^2 - j2\omega ac}{\left[2c^2 - (\omega a)^2\right]^2 + (2\omega ac)^2}\cos\theta \tag{4b}$$

$$\mathsf{F}(A) = \vartheta^{(3)}\frac{U}{\pi^2 - (\omega T_c)^2}\left(e^{-j\omega T_c} + 1\right) \tag{4c}$$

$j$ – imaginary unit and $\vartheta^{(3)} = \pm\pi^2/2$ for sphere-sphere impact and $\vartheta^{(3)} = 1.067\pi^2$ for sphere-slab impact.

The case of sphere-slab impact is treated below. As we know, the nature of hertzian sound is the oscillation of rigid solid and so the source has a character of dipole source, also shown in the Fig. 1b. Hence, the amplitude of an oscillating sphere is dependent on the $\cos\theta$ term and the phase of the acoustic pressure field changes by 180° at $\theta = 90°$, i.e. the rarefaction wave changes into a compression wave or vice-versa. In the case of the sphere-slab impact shown in Fig. 1b, the total pressure field is modeled as the addition between the compression wave and the slab-reflected rarefaction wave of the acoustic dipole. Thus, the addition becomes a subtraction because the reflected rarefaction wave does not shift in phase so there are two waves (compression and rarefaction) arriving to the sensor almost in the same time (Akay and Hodgson, 1978). This phenomenon is also model in acoustics by the method of images or mirrors, where it is considered that there is like a mirror of the impacting particle emitting the rarefaction wave.

The same subtraction is applied in the case of complex spectra to obtain the total spectrum $\mathsf{F}_{im}$, Eq. (5). In this formula, the first term of the right member is attributed to the impacting sphere whereas the second term pertains to the mirror. The time delay $T_d$ of sound arrival due to distance of measurement and sphere's geometry makes that the two terms do not perfectly cancel out or do not arrive in the same time at the sensor.

$$\mathsf{F}_{im}(\omega) = \mathsf{F}_{acc}(\omega) - \mathsf{F}_{acc}(\omega) \cdot e^{-j\omega T_d} \tag{5}$$

Introducing Eq. (2), (3) and (4a-c) in Eq. (5), one obtains the complex magnitude spectrum of the impact between a sphere and a slab. The spectrum contains complex numbers so one applies the multiplication of the spectrum and its conjugate to compute the magnitudes of energy spectrum, Eq. (6).

$$|\mathsf{F}_{im}|^2 = \mathsf{F}_{im} \cdot \mathsf{F}_{im}^* \tag{6}$$

[revised manuscript text omitted]
 such a depth, i.e. 2.5 m, that the cutoff phenomenon is located below 1000 Hz which cannot affect the frequencies associated to the SGN. The pebble-sized particles that are up to 64 mm give SGN of dominating frequencies well above 1000 Hz, whereas the channel's depth of 2.5 m fixes the cutoff frequency to about 148 Hz, assuming a perfect rigid bottom.

15     Therefore, the spectra in the bedload bandwidth will not be exposed to frequency cutoff so this does not present any risk to inversion. Yet, SGN monitoring and inversion technique for GSD determination is particularly adapted to large rivers. Generally, propagation effects are frequency dependent and higher frequency ranges are more affected by attenuation or scattering effects. A solution to the non-linear effects of acoustic propagation would be to determine the river's transfer function by active acoustic experiments (Rigby et al., 2016) and to construct laws of attenuation that will compensate the loss

20     (Wren et al., 2015).

    At first sight, our comparison with (Thorne, 1985, 1986a)'s regression laws would be very naïve due to the nature of theories: we considered the sphere-slab impact whereas the regression laws are from sphere-sphere impact phenomena. Therefore, the inversion is put into discussion when the bed river is no longer armoured and so, the model of impact between sphere and slab is debatable. Here, the target are the large gravel rivers. The dictionaries Δ for both impact models uses an

25     impact velocity $U_{imp} = 1$ ms$^{-1}$, the material is granite and a simulated GSD is used. The GSD is modelled using the (Recking, 2013)'s procedure for which $D_{84} = 2 \cdot D_{50}$, 
[revised manuscript text omitted]

---

## Author Comment (AC2) · 30 Oct 2017

NOTE: The authors did a change in the Eq. (7) which will slightly changes the results of numerical tests and field spectra inversion presented in this article. The change has the physical meaning of passing from power to energy representation of impacts and so it avoids the usage of time in the impact modeling. As the clarity of the presentation was invoked by the two referees, we did some changes regarding the system of notation in order to define easier-to-read equations. In consequence, some parts of the text change but the modifications still follow the pertinent advices of the referees. The resulted paper is also added in the supplement of this response comment.

[Figure]

Below, we are considering the referee's suggestions and corrections step by step.

1. P2/L3: Reference (Parker, 1990) must be mentioned here.

Correction applied. Thanks !

2. Put the citations: "The development of surface-based and mixed-size transport models concerned many scientists (Heimann et al., 2015; Kuhnle, 1993; Parker, 1990; Recking, 2016; Wilcock and Kenworthy, 2002; Wilcock and McArdell, 1993)."

3. Rephrasing :

"Therefore, measuring bedload leads not only to transport rates but also to bedload GSD to calibrate models. However, obtaining bedload samples during exceptional hydraulic events may be difficult by using traditional bedload sampling techniques (e.g., pressure-difference samplers). To measure a wide range of discharge flows, the scientific community has been interested in developing indirect, or surrogate, methods that achieve continuous measurements no matter the hydraulic conditions. This paper is dedicated to the monitoring of bedload GSD using the acoustic noise naturally generated by bedload transport in rivers."

to the following:

"Therefore, measuring bedload leads not only to transport rates but also to bedload GSD to calibrate models (Parker, 2002; Wilcock et al., 2009). However, obtaining bedload samples during exceptional hydraulic events may be difficult by using traditional bedload sampling techniques (e.g., pressure-difference samplers) (Bunte et al., 2010). To measure a wide range of discharge flows, the scientific community has been interested in developing indirect, or surrogate, methods that achieve continuous measurements no matter the hydraulic conditions (Gray et al., 2010; Hubbell, 1964)."

4. The remark ", so-called the bedload Self-Generated Noise (SGN)." is added at the end of the sentence. Thanks!

[Figure]

5. Replacement applied. Thanks!

6. Correction applied

7. Correction is applied:

"Measuring bedload GSD with passive methods has been achieved using plates (Bar-rière et al., 2015; Krein et al., 2014; Rickenmann et al., 2014; Wyss et al., 2016b)".

8. Replacing SGN with "Self-Generated Noise (SGN)"

9. Rephrasing:

"The acoustic effect of accelerating rigid bodies is mathematically modeled by (Kirch-hoff, 1883). A framework was constructed by (Goldsmith, 2003; Hertz, 1882; Hunter, 1957) to mathematically model acceleration profiles from elastic impacts between two solid rigid bodies like two spheres or a sphere and a slab. Mathematically, the acoustic pressure field generated from the acceleration of a rigid body is evaluated by the in-tegral convolution from Eq. (1) (Akay and Hodgson, 1978; Koss and Alfredson, 1973; Thorne and Foden, 1988)"

to the following:

"The acoustic effect of accelerating rigid bodies is physically modeled by (Kirchhoff, 1883). A framework was constructed by (Goldsmith, 2003; Hertz, 1882; Hunter, 1957) to model acceleration profiles from elastic impacts between two solid rigid bodies like two spheres or a sphere and a slab. In a mathematical sense, the acoustic pressure field generated from the acceleration of a rigid body is evaluated by the integral convo-lution from Eq. (1) (Akay and Hodgson, 1978; Koss and Alfredson, 1973; Thorne and Foden, 1988)"

10. All this part containing the term explanation is reformulated according to the new system of notations.

Rephrasing:

"where t is the temporal variable and $\chi$ = t, if 0 ≤ t' ≤ Tc, t' is the delayed time due to sphere geometry, t' = t - (r - a)/c, r – the distance of measurement of the sound from the contact point, see Fig. 1a-b, a – the radius of sphere, c – the sound celerity, $\chi$ = Tc, if t' > Tc, and s – material density and U – the impact velocity. In Eq. (2), the constant ÏŚ(1) = 9.229, for the sphere-sphere impact where the spheres' radii are equal, and ÏŚ(1) = 10.601, for the impact between sphere and slab (considered here). The material parameter $\zeta$ = (1-$\nu$2)/($\pi$E), where E – Young's modulus, $\nu$ – Poisson ratio. The general form of acceleration profile is provided by (Goldsmith, 2003) and it is rewritten in a more convenient form for both impact models in Eq. (3)."

to the following:

"where $\chi$ is the time interval of convolution, with $\chi$ = t, if 0 ≤ $\tau$ ≤ Tc, and $\chi$ = Tc, if $\tau$ > Tc, with $\tau$ a delayed time due to sphere geometry, $\tau$ = t - (r - a)/c, r is the distance between the observation point and the impact, see also the Fig. 1a-b, a is the radius of sphere, c is the sound celerity and s is material density and Uimp is the impact velocity. The parameter ÏŚ(1) is a constant, ÏŚ(1) = 9.229 for the impact between two spheres of same radii and ÏŚ(1) = 10.601, for the impact a slab and a sphere. The parameter $\zeta$ = (1-$\nu$2)/($\pi$Elong) is a material parameter and it contains the Young's modulus (Elong) and the Poisson ratio ($\nu$). The general form of acceleration profile is provided by (Goldsmith, 2003) and it is rewritten in an unified form for both sphere-sphere and sphere-slab impact models, see the Eq. (3)."

11. Acronym explained

12. The verb "is" is inserted. Thanks!

The definition of both frequencies is added at the end of the statement:

"(...) and $\omega$ is the angular frequency which is a measure of rotation rate, in radians per seconds, and it is equal to 2$\pi$f, f is the linear frequency, a measure of number of occurrences per second."

13. The Eq. (7) represents the spectral analytic model of impact between sphere and slab and it is one of the contributions of this paper. In the paper of (Akay & Hodgson, 1978) we only find the temporal analytic model which is discussed in the Appendix 2 and which is used to model the temporal waveform of impact in the Fig. 2a.

14. On P6 Line 25 was given the argument for the use of slab-sphere impact instead of sphere-sphere impact. This means that the slab do not contribute to hertzian sound production as it does not oscillate; his role in the sound production is the reflection of the sound from the impactor particles, by which it allows the method of image modelling. As the impactee particles in the bed river are fixed and usually greater than moving particles, also depicted in the Fig. 1b, we may consider the bed river as a massive slab which reflects the hertzian sound pressure generated by impactor particles. In this way, we skip the task of determining the dimensions of impactee particles.

Rephrasing:

"This could justify the reason to use the sphere-slab impact physics instead of sphere-sphere impact, to reduce eliminate the need for the dimensions of the impactee object"

to the following:

"In this paper, we choose to use a slab model to model bedload SGN as it simplifies the inverse problem. Indeed, the task of determining the dimensions of impacted particles is skipped. Therefore, we consider that the riverbed could be modeled as a slab. This hypothesis could be supported when the riverbed is armored or paved, as in the Fig. 1b, but may be false when the river bed is totally mobile and when the impacts between particles of different diameters are very common."

15. Rephrasing:

"The random variable of the GSD is the number of collisions N, and so the complete notation is $\Gamma$PMF(N = ni). However, one needs to transforms this variable to the weight (mass) of sediments M, to facilitate the comparison with the measured GSD by bedload

samplers. Thus, the variable N from $\Gamma$PMF(N = ni) will be multiplied by Di3 , Eq. (10a) becoming $\Gamma$PMF(M = mi). Furthermore, the solution is written as a cumulative distribution form, $\Gamma$GSD(M $\leq$ mi), expressing the mass percentage of sediments finer than Di, as in Eq. (10b)"

to the following (according to the new system of notations):

"Therefore, the random variable here is probability of collisions $\gamma$, and so the complete notation is $\gamma$(I = Ii) because it is a discrete probability (we operate on size classes of 1-mm diameter). This probability is computed from histogram of number of impacts per second so one needs to transforms it into a histogram in mass of sediments M, to be compatible with the measured GSD by physical sampling. In consequence, $\gamma$(I = Ii) will be scaled by Di3, as in the Eq. (10a) in order to obtain $\gamma$m(M = mi). Finally, the grain size distribution (GSD), or the cumulative distribution form of $\gamma$m, will be $\Gamma$m(M $\leq$ mi), expressing the probability of sediments finer than Di, as defined in the Eq. (10b)."

16. The sentence is completely removed to avoid the ambiguity.

17. The sentence including this line "The distance of measurement is also important, as it the phase shift Td used in the addition of two coherent acoustic fields and also the Poisson's ratio plays a decent role in this variation." is rephrased as it follows:

"The distance of measurement, r, and the material properties, s, $\nu$ and E, play also a role in fpeak variation".

18. Rephrasing :

"The computation of high order analysis could be made by Sobol's analysis (Sobol, 2001), but this type of analysis is beyond the scope of this article"

to the following:

"The computation of high order sensitivity indices can be made using Sobol's methodology (Sobol, 2001), but this type of analysis is beyond the scope of this article".

19. Replacement applied. Thanks!

20. As with the new Eq. (7), the values of sensitivity indices obtained by FAST analysis change a bit.

21. This part will be treated in the Discussion part of the paper (see also the referee's point #37 )

22. Replaced "LS" with "Least Square (LS)"

23. Figure 2c added with modeled spectra

24. Replaced "solution" with "least square solution"

25. This part will describe the PSD formulation for stationary signals and energy formulation for energy signals

Rephrased from:

"A particular concern in the theory of statistical signal processing is the variance of computed PSD. In our work, Short-Time Fourier Transform has been used. Fourier transforms are applied on small temporal windows of signal (with an overlap of 50 %). These collections of local spectra are averaged over predefined frequency bins which are narrowband (Oppenheim and Verghese, 2010). If the signal is long enough then the averaging of a great number of local spectra diminishes the spectrum variance. In the same time a good spectral resolution is achieved and a great number of size classes may be correctly inferred from the PSD."

to the following:

"The signal processing tools in this paper refers to using the Power Spectral Density (PSD) as the method of spectral representation of bedload signal. The use of PSD is worthwhile because the type of bedload signal is a stationary random one. Random stationary signals are signals varying in time but whose average and standard deviation of amplitude values over some fixed periods are constant. A particular concern

for the signal processing of random processes is the minimization of the variance on the PSD. In our work, we will use the periodogram approach of PSD estimation, which means applying the Fourier transform on local portions (windows) of random signal, with an overlap of 50%, and then averaging them in narrow bandwidths (Oppenheim and Verghese, 2010). Therefore, the averaging is useful because it mitigates the variance on the PSD. In this work, the quality of spectra is vital for accuracy of estimations. The uncertainty principle tells us that the smaller the temporal window, the bigger the uncertainty in locating two very close frequencies on the spectrum, so a trade must be made between the PSD variance and its spectral resolution. If the bedload signal is too short, the quality of spectra toward the low frequency bands is worsened because in one single bandwidth of the Fourier Transform there are spectral information of impacts from multiples grain sizes. Finally, the longer the signal the better the spectral resolution and the lesser the variance on the PSD curve."

26. Replacement applied

27. Rephrasing:

"The presented PSD curve shows two main bandlimited phenomena: (1) from 10 to 1000 Hz, which does not represent the bedload process but hydrodynamic processes; (2) from 1000 Hz to 50000 Hz which truly represents the bedload transport. The inversion procedure to estimate the GSD will be reliable as long as the bedload bandlimited region does not interfere with other extraneous noise source (hydrodynamic noise, turbulence)."

to the following:

"Two main sources of noise can be distinguished in the recordings: below and above 400 Hz (fig. 7). Bedload impacts can clearly be heard in the higher frequency band, it sounds like the crackling of the flames. Sounds occurring below 400 Hz are not propagating sounds as they are localized below the cutoff frequency of the river waveguide (Geay 2017b, Rigby 2016). They are related to turbulence induced noise around the

sensor and to mechanical movements of the structure sharing the hydrophone.".

28. We do not consider sparse grain size distributions, i.e. logarithmic size classes like the Wenthworth definition. The grain size distributions estimated in this paper have the fixed resolution of 1 mm and start from 1 mm (coarse sand).

29. Correction applied

30. The sentence containing the word "noisiest" is reformulated as it follows:

from

"Moreover, the most sediments were sampled in this position and the SPL map from Figure 7c showed that this position is the noisiest from all the cross-section."

to

"In this position, it can be observed that a maximum bedload acoustic energy has been recorded. Additionally, a maximum flux of sediments was sampled in this position."

31. Replacement applied

32. Replaced "a certain value" with "10 dB"

33. Modified as in the point #14 of RC 1: "1-2 mm" to "10-14 mm". Thanks.

34. Removed "observed to be"

35. Rephrasing:

"The cross-sectional variation of the estimated D16, D50 and D84 by the NNLS algorithm follows quite decently the trend of bedload D50 measured by the TR sampler, Fig. 9b."

to the following:

"The cross-sectional variation of the estimated D16, D50 and D84 by the NNLS algorithm follows the same trend of increasing values from left to right banks as the bedload

D50 measured by the TR sampler, Fig. 9b. However, the cross-sectional variability of sampled diameters is higher than the estimated one".

36. Rephrasing:

"The model Eq. (9) is valid if the acoustic propagation only takes into consideration the sound divergence models."

to the following:

"The proposed model (eq. 9) has been elaborating by assuming a simple geometrical spreading model of the acoustic waves in the river."

37. Discussion based on acoustic propagation study by (Geay et al.,2017b):

Rephrasing (the text from P15-L3 to P15-L15):

"Thus, the acoustic propagation has effects on the spectral amplitudes but not on the spectrum's shape. In nature, however, there are many other acoustic propagation models in the river. One of them is when the high frequencies are more attenuated than the lower ones. Also, higher frequencies are prone to scattering effects or to absorption, discussed in the context of river 5 soundscape in (Tonolla et al., 2009). In the case of Isère Rivers, even though the suspended sediment transport is important, these effects are assumed to be mitigated by the fact the sound production from the powerful acoustic source from the centre overtakes assures enough good SNR of recorded signal. Another propagation effect concerns the lower frequencies, which are attenuated by the frequency cutoff phenomena, due to acoustic propagation in shallow waters, or waveguides (Geay, 2013; Jensen et al., 2011; Rigby et al., 2016). During experimental fields, the Isère River has enough depth, 2.5 m, that the cutoff 10 phenomenon cannot affect the generate frequencies associated to SGN. The pebble-sized particles that are up to 64 mm give SGN of dominating frequencies well above 1000 Hz, whereas the channel's depth of 2.5 m fixes the cutoff frequency to about 148 Hz, assuming a perfect rigid bottom. Therefore, the spectra in the bedload bandwidth will not be exposed to

frequency cutoff so this does not present any risk to inversion. Yet, SGN monitoring and inversion technique for GSD determination is particularly adapted to large rivers."

to the following:

"Bedload SGN spectra monitored by a hydrophone are not only dependent on bedload sizes but also affected by propagation effects. For example, an alpine river has been modelled as a Pekeris waveguide (Geay et al., 2017). Consequently, it has been shown that the monitored spectra were slightly dependent on the hydrophone position in the lower frequency band.Another propagation effect concerns the frequency cutoff phenomena, due to acoustic propagation in waveguides (Geay, 2013; Geay et al. 2017b; Jensen et al., 2011; Rigby et al., 2016). During experimental fields, the Isère River such a depth, i.e. 2.5 m, that the cutoff phenomenon is located below 1000 Hz which cannot affect the frequencies associated to the SGN. The pebble-sized particles that are up to 64 mm give SGN of dominating frequencies well above 1000 Hz, whereas the channel's depth of 2.5 m fixes the cutoff frequency to about 148 Hz, assuming a perfect rigid bottom. Therefore, the spectra in the bedload bandwidth will not be exposed to frequency cutoff so this does not present any risk to inversion. Yet, SGN monitoring and inversion technique for GSD determination is particularly adapted to large rivers. Generally, propagation effects are frequency dependent and higher frequency ranges are more affected by attenuation or scattering effects. A solution to the non-linear effects of acoustic propagation would be to determine the river's transfer function by active acoustic experiments (Rigby et al., 2016) and to construct laws of attenuation that will compensate the loss (Wren et al., 2015)."

38. The last sentence from P16/L29 is entirely removed

39. Tables and Figures: Overall, tables and figures support well the text. Event though not critical, legend and axis labels are too small and the arrangement of subfigures is not well balanced for most of the figures. Fig. 6c- Please consider to set the background color as white, as done for Fig. 6a. You could also report the cross-section on

this panel and keep the same y-axis direction between Fig. 6a and 6c. Two "in" in the caption ("in in units of dB. . ."). Fig. 10- replace "medallion" by "inset". The title of table 3 is not enough explicit.

Captions revisited and replacement applied. Thank you !